**Investigation**

# Genotype-by-environment interactions shape ubiquitin-proteasome system activity

Randi R. Avery (ID) , Mahlon A. Collins,*,[†] Frank W. Albert (ID) *,[†]

Department of Genetics, Cell Biology, and Development, University of Minnesota Twin Cities,  Minneapolis, MN 55455, United States

*Corresponding authors: Mahlon A. Collins, Department of Genetics, Cell Biology, and Development, University of Minnesota Twin Cities, 6-160 Jackson Hall, 321 Church St SE, Minneapolis, MN 55455, United States. Email: mahlon@umn.edu; Frank W. Albert, Department of Genetics, Cell Biology, and Development, University of Minnesota Twin Cities, 6-160 Jackson Hall, 321 Church St SE, Minneapolis, MN 55455, United States. Email: falbert@umn.edu
[†]These authors jointly supervised this work.

In genotype-by-environment interactions (GxE), the effect of a genetic variant on a trait depends on the environment. GxE influences numerous organismal traits. However, we have limited understanding of how GxE shapes molecular processes. Here, we characterized how GxE shapes protein degradation, an essential molecular process that affects cellular and organismal physiology. Using 2 isolates of the yeast Saccharomyces cerevisiae, we profiled GxE in the ubiquitin-proteasome system (UPS), the primary protein degradation system in eukaryotes. By measuring UPS degradation activity toward 6 substrates that engage multiple distinct UPS pathways across 8 diverse environments, we discovered extensive GxE in the genetics of the UPS. The effects of all environments, including environments previously reported to affect UPS activity, differed between isolates and UPS substrates. To identify genomic regions underlying GxE for UPS activity, we mapped genetic influences on all our environment–UPS substrate combinations. Hundreds of locus effects varied depending on the environment. Most of these corresponded to loci that were present in one environment but not another ("presence/absence" GxE), while a smaller number of loci had opposing effects in different environments ("sign change" GxE). The number, genomic location, and type of GxE (presence/absence or sign change) of loci exhibiting GxE varied across UPS substrates. Loci exhibiting GxE were clustered at genomic regions that contain core UPS genes and at regions containing variation that affects the expression of thousands of genes, suggesting indirect contributions to UPS activity. Our results reveal complex interactions between the environment and the genetics of protein degradation.

Keywords: genotype-by-environment interactions; ubiquitin-proteasome system; quantitative trait loci; Saccharomyces cerevisiae

## Introduction

Genotype-by-environment interactions (GxE) occur when a genetic variant's effect on a trait is modulated by the environment. GxE can have profound effects on organismal physiology. For example, in the disease phenylketonuria, individuals with 2 defective copies of the phenylalanine hydroxylase gene develop severe symptoms, including brain damage and intellectual disabilities, if they consume a diet with standard amounts of phenylalanine. However, most symptoms can be avoided by consuming a diet with reduced phenylalanine (Bickel et al. 1953; Guthrie 1961; Shostak 2003; Widaman 2009). Other prominent examples of GxE exist in pharmacogenetics, where genetic differences modulate drug efficacy (Pirmohamed 2023). Thus, understanding the extent and genetic basis of GxE has been a longstanding goal in biomedical research.

Efforts to this end have revealed that GxE is widespread at the level of organismal traits. GxE has been observed for a variety of morphological (eg *Drosophila melanogaster* bristle number; Gurganus et al. 1998) and developmental (eg flowering time in *Arabidopsis thaliana* at various temperatures; Sasaki et al. 2015) traits in numerous organisms. Recent work in humans has begun to explore the impact of environmental factors on traits related to health and disease. By using self-reported and demographic

information to integrate environmental factors into genome-wide association studies (GWAS), these efforts have revealed that GxE shapes the genetics of a variety of clinical syndromes, including depression (Li et al. 2022), cancer (Yang et al. 2020), and health-related traits, such as body mass index (Robinson et al. 2017).

However, to what extent GxE occurs at the level of the molecular processes that give rise to organismal phenotypes in humans and other species is poorly understood. A key challenge is that most traits are genetically complex, influenced by variation at loci throughout the genome. Profiling sufficiently large samples to achieve the statistical power needed to detect the effects of multiple loci and their interaction with environmental factors requires assays with high throughput and quantitative precision. This has limited our ability to measure how environmental factors modulate genetic influences on all but a small number of molecular processes. Notably, studies of gene expression have shown that molecular processes also exhibit widespread GxE (Grishkevich and Yanai 2013; Boye et al. 2024). For example, studies in flies (Huang et al. 2020), plants (Cubillos et al. 2014), roundworms (Li et al. 2006), and mice (Ballinger et al. 2023) have assayed gene expression in genetically different individuals at different temperatures. In these studies, GxE predominantly

occurred at loci that influence gene expression ("expression quantitative trait loci," eQTLs) via *trans*-acting mechanisms. In humans, eQTLs identified in immune cells display considerable GxE, including eQTLs at immune genes whose effects become detectable only when the given immune pathway is activated, demonstrating that GxE can occur via direct effects on genes in a relevant pathway (Fairfax et al. 2014; Nédélec et al. 2016; Quach et al. 2016; Kim-Hellmuth et al. 2017). Profiling stimulated human immune cells has also revealed stimulation-dependent *trans*-eQTLs, showing that GxE can occur via indirect mechanisms (Fairfax et al. 2014; Lee et al. 2014). These and other context-specific eQTLs displaying GxE are enriched for GWAS signals for complex organismal traits (Kim-Hellmuth et al. 2017; Lea et al. 2022), highlighting the value of studying molecular traits that give rise to organismal phenotypes.

The yeast *Saccharomyces cerevisiae* has served as a powerful model for dissecting GxE (Yadav and Sinha 2018) because thousands of natural, genetically different isolates (Warringer et al. 2011), their cross progeny (Smith and Kruglyak 2008; Bloom et al. 2019; Nguyen Ba et al. 2022), or strains harboring engineered natural variants (Chen et al. 2023) can be exposed to tightly controlled environments at high levels of replication. These approaches have revealed widespread GxE in yeast growth. A survey of natural yeast strains showed that isolates from genetically different populations grew differently in nearly half of 200 assayed environments (Warringer et al. 2011). Later work using linkage mapping in crosses revealed considerable heterogeneity in the genetic architecture of growth in dozens of environments, including loci that only affected growth in specific environments and loci whose direction of effect differed between environments (Bloom et al. 2013; Gagneur et al. 2013; Nguyen Ba et al. 2022). Recently, Chen et al. (2023) revealed that 93.7% of natural variants that had a significant effect on growth in at least one condition showed evidence of GxE. Additionally, in a cross between 2 yeast strains, Smith and Kruglyak (2008) found GxE at about 40% of loci that shape transcript abundance in media with glucose versus ethanol as the carbon source.

Despite these insights from humans and model systems, our knowledge about GxE in molecular traits remains limited. Profiling additional molecular processes with known roles in organismal physiology would expand our understanding of how GxE shapes health, disease, and evolution. Protein degradation is an essential molecular process that influences numerous aspects of cellular and organismal physiology. In eukaryotes, most protein degradation (70% to 80%) is carried out by the ubiquitin-proteasome system (UPS) (Bachmair et al. 1986; Coux et al. 1996; Hershko and Ciechanover 1998; Collins and Goldberg 2017). By degrading substrate proteins, the UPS regulates protein abundance and removes misfolded and damaged proteins from cells (Hanna and Finley 2007; Varshavsky 2011; Collins and Goldberg 2017). The central importance of UPS protein degradation is illustrated by defects in this process that occur in numerous human diseases, including cancers, immune disorders, and neurodegenerative diseases (Schwartz and Ciechanover 1999; Shringarpure and Davies 2002; Dantuma and Bott 2014; Zheng et al. 2014). The UPS comprises the ubiquitin system, a collection of enzymes that mark substrate proteins for degradation, and the proteasome, a multi-protein complex that degrades substrate proteins to small peptides. The ubiquitin system recognizes short signal sequences (termed "degrons"; Varshavsky 1991) in proteins, and then marks the substrate protein for degradation by covalently attaching the small protein ubiquitin. Ubiquitinated substrate proteins are bound by the proteasome's 19S regulatory particle,

unfolded, and degraded to short peptides by the proteasome's 20S core particle (Hershko and Ciechanover 1998; Finley et al. 2012; Bett 2016). The proteasome can also bind and degrade certain substrates directly, independent of the ubiquitin system (Finley et al. 2012; Ben-Nissan and Sharon 2014). UPS protein degradation is organized into multiple distinct pathways based on the ubiquitin system enzymes and proteasome receptors involved in targeting and binding substrates of a given pathway. This can result in highly pathway- and even substrate-specific UPS activity that is tailored to the physiological needs of the cell. For example, during proteotoxic stress, UPS activity toward misfolded proteins can selectively increase (Gardner et al. 2005; Rosenbaum et al. 2011; Ibarra et al. 2016).

We recently showed that UPS activity is a genetically complex trait (Collins et al. 2022, 2023). By measuring UPS activity toward multiple substrates that engage distinct UPS targeting and degradation pathways, we revealed that many variant effects are substrate-specific in that the magnitude and, in some cases, direction of their effects on UPS activity varied between substrates. Some loci exerting substrate-specific effects contained causal variants in UPS genes, while other loci did not contain any genes with known roles in UPS activity, suggesting indirect effects (Collins et al. 2022, 2023). However, these experiments were performed in a single environment and thus, the extent of GxE in the genetics of UPS activity is unknown.

Protein degradation is highly environment-dependent, raising the possibility that the genetics of UPS activity is subject to GxE. For example, in environmental conditions that cause misfolded or oxidatively damaged proteins to accumulate, UPS activity increases to clear these molecules from the cell (Grimm et al. 2012; Sontag et al. 2014; Finley and Prado 2020). In contrast, in nutrient-poor conditions, UPS protein degradation, an ATP-dependent process, decreases to conserve cellular resources (Bajorek et al. 2003; Laporte et al. 2008; Waite et al. 2016). This process is well-characterized in yeast cells, which decrease UPS activity in low-glucose environments by sequestering proteasomes in inactive aggregates and in low nitrogen environments by autophagically degrading proteasomes (Laporte et al. 2008; Li et al. 2019). These environmental influences on UPS activity, combined with our prior observation that UPS activity is affected by complex natural genetic variation, suggest that genetic influences on UPS activity may be influenced by GxE.

Here, we quantified and mapped GxE in the genetics of UPS activity. By measuring genetic influences on UPS activity toward 6 substrates that engage distinct UPS pathways in 8 environments, we discovered extensive GxE in UPS activity between 2 genetically different yeast strains. Genetic mapping revealed hundreds of instances where a locus altered UPS activity in one environment but not another, as well as a smaller number of instances where a locus effect changed direction between environments. Our results reveal a high degree of GxE in the genetic architecture of UPS activity and expand our understanding of how GxE shapes molecular traits.

## Methods
### UPS activity reporters
To measure UPS activity, we used tandem fluorescent protein timers (TFTs) (Khmelinskii et al. 2012), a 2-color fluorescent reporter system that provides high-throughput measurements of protein turnover. TFTs are linear fusions of 2 fluorescent proteins. In the most common implementation, the TFT consists of a faster-maturing green fluorescent protein (GFP) and a more slowly

maturing red fluorescent protein (RFP). Because the 2 fluorophores fold and emit fluorescence over distinct time scales, the ratio of RFP to GFP changes over time. If the TFT's degradation rate is faster than the RFP's maturation rate, the TFT ratio can be used to measure the construct's degradation rate (Khmelinskii et al. 2012; Khmelinskii and Knop 2014). We used superfolder GFP (sfGFP) as the GFP in all TFTs and mCherry or mRuby as the RFP as indicated below (Supplementary Table 2). As in prior studies, we inserted an unstructured 35 amino acid sequence (GSGSREARHKQKIVAPV KQTLNFDLLKLAGDVESN) between the fluorophores to minimize fluorescence resonance energy transfer (Khmelinskii et al. 2012; Collins et al. 2022, 2023).

To relate the output of our TFTs to UPS activity, we designed reporters to include degrons that engage distinct UPS pathways. The resulting constructs provide quantitative, high-throughput, substrate-specific readouts of UPS activity in live, single cells that are sensitive to genetic and chemical perturbations that alter UPS activity (Collins et al. 2022, 2023). We measured the activity of 4 ubiquitin system-dependent UPS pathways. Three of these engage distinct branches of UPS N-degron pathways, in which a protein's N-terminal amino acid functions as a degron (hereafter, an "N-degron"; Varshavsky 2019). Our N-degron reporters include the threonine N-degron of the Ac/N-degron pathway ("Thr"), the asparagine N-degron of the type I Arg/N-degron pathway ("Asn"), and the phenylalanine N-degron of the type II Arg/N-degron pathway ("Phe"). The fourth ubiquitin system-dependent reporter contained a noncleavable ubiquitin moiety (ubiquitin G76V) as its degron at the N-terminus of the TFT ("UFD"). The resulting construct measures the activity of the ubiquitin fusion domain pathway, a UPS pathway involved in protein quality control (Johnson et al. 1995; Theodoraki et al. 2012; Devarajan et al. 2020). We also created 2 reporters that measure the activity of ubiquitin system-independent UPS pathways. One reporter contains the ubiquitin-independent degron encoded in the 80 N-terminal amino acids of the Rpn4 protein (hereafter, "Rpn4"). The second reporter contains a linear chain of 4 ubiquitin molecules (hereafter, "4xUb"). Each of the ubiquitins in 4xUb have the G76V substitution so they cannot be cleaved, and K29/48/63R substitutions to prevent further ubiquitination. Both the Rpn4 and 4xUb reporters are directly bound and degraded by the proteasome without being targeted by the ubiquitin system. Their activities thus provide a readout of proteasome activity independent of the ubiquitin system. However, they are each bound by distinct proteasome receptors.

Based on the half-lives of the N-degrons (Bachmair et al. 1986; Varshavsky 2011), we used mCherry as the RFP for all reporters except the Thr N-degron reporter. For Thr, we used mRuby, which matures over a longer time scale than mCherry (168 vs. 40 min; Shaner et al. 2004; Kredel et al. 2009) to improve its dynamic range. The 4xUb and UFD reporters were engineered for this study using procedures described in Collins et al. (2022, 2023). We packaged each reporter into plasmid backbone BFA0190 (Collins et al. 2022, 2023) containing common sequence elements for reporter integration, selection, and expression. Each reporter contains the TDH3 promoter to drive strong, constitutive TFT expression, the ADH1 terminator, codon-optimized sfGFP and mCherry or mRuby, and a KanMX cassette to select for the presence of the reporter via resistance to the antibiotic G418. These elements are flanked by sequences homologous to the genomic regions immediately up- and downstream of LYP1. Transformation of the reporter containing these flanking sequences results in integration at the LYP1 locus, which can be selected for using the toxic amino acid analogue thialysine. DNA fragments of the reporters used for

transformations were made by PCR amplifying the sequence on the plasmid carrying the reporter sequence (Supplementary Table 8). The PCR fragments were purified using Monarch PCR & DNA Cleanup Kit (5 μg) from New England Biolabs (NEB) Cat#T1030L, or ran on an electrophoresis gel and purified using the Monarch DNA Gel Extraction Kit (NEB) Cat#T1020L, according to the manufacturer's protocol.

## Yeast strains and handling

All experiments used yeast strains derived from 2 genetically divergent *S. cerevisiae* strains. The haploid BY strain (genotype: *MAT**a** his3Δ hoΔ*) is closely related to the S288C laboratory strain. The haploid RM strain (genotype: *MATα can1Δ::STE2pr-SpHIS5 his3Δ::NatMX AMN1-BY hoΔ::HphMX URA3-FY*) is derived from a wild strain that was originally isolated from a California vineyard. To characterize UPS reporters, we also used a previously characterized BY strain lacking the RPN4 gene (genotype: *MAT**a** his3Δ hoΔ rpn4Δ::NatMX*) (Collins et al. 2022). All strains used in the present study are listed in Supplementary Table 9.

We built strains harboring our UPS activity reporters using the following procedures. Transformations using the reporter sequence fragments were performed using the Zymo Frozen-EZ Yeast Transformation II Kit Cat#T2001 according to the manufacturer's protocol or the lithium acetate/single-stranded carrier DNA/poly-ethylene glycol method (Gietz and Schiestl 2007) as described in Collins et al. (2022, 2023). We verified the presence of the reporters in the desired locus with colony PCR. Eight confirmed transformants for each reporter in each strain were collected as independent biological replicates.

## Yeast mating and segregant populations

To create large, genetically diverse cell populations for genetic mapping, we used a previously described approach (Ehrenreich et al. 2010; Albert et al. 2014; Brion et al. 2020). We created segregant populations containing each UPS activity reporter using a modified synthetic genetic array methodology (Baryshnikova et al. 2010; Kuzmin et al. 2016). Briefly, BY strains (*MAT**a***) containing each reporter were mixed with a wild-type RM strain (*MATα*, without a reporter: YFA0039) on solid YPD medium (all media are described in Supplementary Table 10) and grown overnight at 30 °C. This was done independently for 2 biological replicates based on 2 different colony PCR-confirmed transformants for each reporter. Diploids from the mating were selected on YPD plates containing G418 and CloNAT to select for the UPS reporter in the BY strain and *his3Δ::NatMX* in the RM strain, respectively. Five milliliters of liquid YPD were inoculated with the diploids and grown overnight to saturation at 30 °C with rolling. The diploids were then spun down in 15 mL tubes in a tabletop centrifuge at 3,000 rpm for 2 min. The cell pellet was resuspended in 5 mL of sporulation medium and transferred to glass tubes and incubated at room temperature for 10 d on a turning wheel. We evaluated the extent of sporulation in each culture using brightfield microscopy. When the cultures reached ~80% sporulation, we harvested the spores. To separate the spores from their asci, we spun the spores for 1.5 min at 5,000 rpm in a tabletop centrifuge, discarded the supernatant, resuspended in water with 1 mg/mL Zymolyase lytic enzyme (United States Biological, Salem, MA, USA), and incubated for 2 h, vortexing every half hour. We again washed the cells and plated them onto solid haploid selection medium with G418 and thialysine. We used this medium to select for recombined haploid cells ("segregants") that contain the reporter via G418, the *MAT**a*** mating type locus via the *Schizosaccharomyces pombe HIS5* gene under the control of the

STE2 promoter (which is only active in MAT**a** cells), and replacement of the LYP1 gene by the reporter via resistance to thialysine. We grew the resulting segregant populations for 2 d on haploid selection plates at 30 °C, harvested the cells from the plates, and stored each population as a separate glycerol stock. We saved 2 biological replicates for each reporter as separate, individual stocks.

## Environments

To characterize how genetic influences on the UPS are shaped by environmental factors, we measured UPS activity in 8 distinct media formulations, which we term "environments" (Supplementary Table 1). As a baseline environment, we used synthetic complete (SC), a nutrient-rich medium with glucose, nitrogen, and amino acids. G418 (200 mg/mL) was added to all environments to maintain the reporter sequence in the genome. We compared UPS activity in SC (SC -His -Lys + YNB + 0.1% MSG + 2% glucose) with that in 7 different environments: low glucose (SC -His -Lys + YNB + 0.1% MSG + 0.025% glucose), low nitrogen (YNB + 2% glucose), yeast nitrogen base (YNB + 0.1% MSG + 2% glucose), SC + 4NQO (4NQO; 2 µg/mL), SC + L-azetidine-2-carboxylic acid (AZC; 4 mM), SC + bortezomib (BTZ; 40 µM), and SC + lithium acetate (LiAc; 20 mM). The media and chemical formulations and concentrations used for all experiments are described in Supplementary Table 10.

## Growth and environmental exposures prior to flow cytometry

Eight biological replicates of each of the BY, RM, and BY *rpn4Δ* strains containing the reporters (Supplementary Table 9) were grown in 96-well plates and incubated at 30 °C on a MixMate (Eppendorf, Hamburg, Germany) at 1,100 rpm. G418 (200 mg/mL) was added to all media except for the negative controls. The incubation times in each medium were determined based on a combination of previous literature (Burgis and Samson 2007; Laporte et al. 2008; Marshall et al. 2016; Waite et al. 2016; Li et al. 2019; Work and Brandman 2021) and preliminary growth rate measurements in a plate reader to ensure that all cultures had similar optical density (O.D.) for flow cytometry and fluorescence-activated cell sorting (FACS).

First, all cultures were grown overnight to saturation in SC medium. Next, from a common saturated culture in SC for each replicate, 4 µL was used to inoculate 400 µL media for each environment. For the SC, LiAc, and YNB environments, 400 µL of the respective medium was inoculated with 4 µL of the overnight growth and incubated for 3 h prior to flow cytometry. For the BTZ environment, 400 µL of SC + BTZ medium was inoculated with 4 µL of the overnight growth culture and incubated for 4 h until flow cytometry measurements. Based on preliminary experiments, cells did not grow well in low nitrogen, low glucose, 4NQO, and AZC. Therefore, after overnight growth in SC, we continued growing the replicates for these 4 environments in SC before exposing the cells to that environment: 4 µL of the overnight culture was added to 400 µL of SC and grown for 3 h. After those 3 h, samples to be exposed to low nitrogen and low glucose were spun down for 5 min at 3,000 rpm in 96-well plates on a tabletop centrifuge. The SC medium was replaced with either the low nitrogen or low-glucose media and incubated 24 h until measured via flow cytometry. If the low nitrogen or low-glucose samples were too dense for flow cytometry, they were diluted in a 1:3 ratio with the same media type. For the 4NQO samples, after the 3 h of growth in SC, the cells were spun down for 5 min at 3,000 rpm in 96-well plates on a tabletop centrifuge. The SC

medium was replaced with SC + 4NQO medium and incubated for 1 h until measured via flow cytometry. For the AZC samples, after the 3 h of growth in SC, 3.2 µL of AZC stock was added to each culture. After 5 h of incubation in AZC, samples were measured via flow cytometry.

We used measurements of the wild-type BY strain (YFA0040) without a TFT reporter grown in SC only to determine background fluorescence levels: 400 µL of SC -lys was inoculated with 4 µL of the overnight growth of YFA0040 and incubated for 3 h prior to flow cytometry. In the experiments characterizing the 4xUb and UFD reporters in SC, 4 of the UFD BY *rpn4Δ* replicates did not produce GFP fluorescence above YFA0040. We therefore excluded those 4 samples without detectable GFP from analysis (Fig. 1c). Flow cytometry samples excluded from analysis of GxE between the BY and RM strains (Fig. 2) are described below.

## Flow cytometry

All flow cytometry experiments were performed on the BD FACSymphony A3 flow cytometer (BD, Franklin Lakes, NJ, USA) at the University of Minnesota University Flow Cytometry Resource. The cytometer is equipped with a 20 mW 488 nm laser with a 488/10 filter to measure forward scatter (FSC) and side scatter (SSC) and a 525/50 filter to measure GFP fluorescence, and a 40 mW 561 nm laser with a 610/20 filter to measure RFP fluorescence. The voltages for each parameter are listed in Supplementary Table 11. We altered voltages for samples containing AZC compared with the other environments so that GFP did not saturate at the upper end of detection, as GFP fluorescence was higher in those samples, as follows: FSC, 450; GFP, 400; RFP, 600.

We used flow cytometry to characterize the 2 new reporters, 4xUb and UFD, in SC medium. We recorded data for 10,000 cells from each of the 8 biological replicates of each strain (BY, RM, BY *rpn4Δ*) containing these reporters (Supplementary Table 9). We used flow cytometry to test for GxE between the BY and RM strains for all 6 reporters. For these experiments, we recorded data for 20,000 cells from the 8 biological replicates of BY and RM strains containing the 6 reporters (Supplementary Table 9) in each of 8 environments as described above. Two 96-well plates for each reporter were used to hold all samples. For the 4xUb and UFD reporters, each of the 2 plates had a set of baseline samples (SC) (Fig. 2a and b).

## Analysis of flow cytometry data

Analyses were conducted using code adapted from Collins et al. (2022). Briefly, we analyzed flow cytometry data using the R (R Foundation for Statistical Computing, Vienna, Austria) package flowCore (Hahne et al. 2009). We first filtered each replicate to include only cells within +/−10% the FSC median (proxy for cell size). This removed cellular debris, aggregates of multiple cells, and restricted our analyses to cells of the same approximate size. The low nitrogen samples showed 2 FSC peaks, perhaps due to incomplete budding of daughter cells. In order to only analyze single cells, we selected the smaller of the 2 peaks for analysis of the low nitrogen samples. The median FSC values of the smaller low nitrogen peaks were similar in size to the FSC medians of all other samples.

As in prior studies (Brion et al. 2020; Collins et al. 2023), we observed that the $-\log_2$(RFP/GFP) ratio changed over time within some replicates of the same strain, reporter, and environment. To correct for this, we used the residuals of a loess regression of the $-\log_2$(RFP/GFP) ratio, as in Brion et al. (2020) and Collins et al. (2023). We refer to the time-corrected $-\log_2$(RFP/GFP) ratio

as "UPS activity" throughout. P-values for differences in UPS activity between the BY, RM, and BY *rpn4*Δ strains were calculated using a 2-tailed *t*-test.

The GxE effect of each reporter/environment combination was determined using the following linear mixed model:

$$model = UPS\ activity \sim strain \times environment + (1|replicate)$$

Here, the random effect term "(1|replicate)" accounts for interindividual variation among independent biological replicates. We conducted pairwise analyses, in which we compared the effect of each of the 7 environments to the baseline SC environment, for 1 reporter at a time. If one of the 8 replicates for a given reporter in a given environment had a median GFP level below that of the negative control (a BY strain with no reporter, YFA0040), we concluded that the UPS activity measured by the associated reporter could not be accurately measured in that environment, and all 8 replicates for that reporter/environment combination were excluded. This exclusion applied to the 4xUb and UFD reporters in low nitrogen, resulting in 40 tests for GxE using the above model. Statistical significance of main effects of strain, environment, and the GxE interaction term between strain and environment was assessed using an ANOVA. We used Bonferroni-corrected P-value thresholds of 0.05/40 = 0.00125. The magnitude of the environmental effect on UPS activity (Fig. 2c and d) was determined by subtracting the mean UPS activity in the given environment from the mean UPS activity in SC, separately for each strain. For the 4xUb and UFD reporters, each of the 2 plates had a set of baseline (SC) samples and these corresponding SC samples were used to calculate "Environment effect" as seen in Fig. 2. To describe GxE patterns revealed by the linear models, we also used *t*-tests comparing UPS activity for each strain between SC and another environment. For these *t*-tests, we used Bonferroni-corrected P-value thresholds of 0.05/80 = 0.000625.

## Growth and environmental exposures prior to FACS

All incubations were performed at 30 °C in glass test tubes with rolling. Two independent biological replicate segregant stocks were thawed and used to inoculate our baseline medium (SC -his -lys + G418) and grown overnight to saturation. This common culture was used to inoculate media for each environment as follows.

For SC, LiAc, and YNB, 4.2 mL of media was inoculated with 800 µL of the overnight growth culture and incubated for 3 h until FACS. For the BTZ samples, 4.5 mL of SC + BTZ medium was inoculated with 500 µL of the overnight growth culture and incubated for 4 h until FACS.

For samples that were to be exposed to low nitrogen, low glucose, 4NQO, and AZC, 500 µL of the overnight SC culture was added to 4.5 mL of SC and grown for 3 h. Samples exposed to low nitrogen and low glucose were then spun down for 2 min at 3,000 rpm in 15 mL tubes on a tabletop centrifuge. The SC medium was replaced with 5 mL of the low nitrogen or low-glucose media and incubated for 24 h until cell sorting. For the 4NQO samples, after the 3 h of growth in SC, 0.4 µL of 4NQO stock was added to the cultures. The 4NQO cultures were immediately vortexed and incubated for 1 h until cell sorting. For the AZC samples, after the 3 h of growth in SC, 40 µL of AZC stock was added to each sample and vortexed. AZC samples were incubated for 5 h until FACS. These volumes and incubation times led to the samples being at approximately the same O.D. when sorted.

A BY strain without a UPS reporter (YFA0040) was grown overnight to saturation in SC -lys to be used as a negative fluorescence control. A 4.5 mL volume of SC -lys was inoculated with 500 µL of the overnight growth and incubated for 3 h until FACS. The segregants with the 4xUb reporter in the low nitrogen environment did not produce GFP fluorescence above the negative control during the FACS experiments, and therefore, no cells were collected for that combination, and all downstream analyses do not include 4xUb in low nitrogen.

## Fluorescence-activated cell sorting

We used FACS to isolate phenotypically extreme cell populations as part of a bulk segregant analysis genetic mapping approach (Albert et al. 2014; Brion et al. 2020). All cell sorting was performed on a FACSAria II cell sorter (BD) by operators at the University of Minnesota Flow Cytometry Resource. To remove doublets from each sample, we used plots of SSC height by width and FSC height by width. We kept cells within the peak of FSC area +/−7.5%, which maintained our primary haploid cell population and excluded cellular debris and aggregates (Collins et al. 2022, 2023). We restricted our sorts to populations of cells with GFP fluorescence above that of the negative control BY strain YFA0040, which does not express any fluorescent proteins. We collected populations of cells from the 2% high and low tails of the RFP/GFP ratio distribution. We aimed to collect pools of 20,000 cells for each of these populations. When cultures did not contain a sufficient amount of GFP-positive cells to collect 20,000 cells, we collected fewer cells. We empirically determined reporter/environment combinations for which the cell pools did not grow well after sorting in a preliminary experiment. We therefore collected more cells for those reporter/environment combinations up to 100,000 cells. The final numbers of cells collected for both replicates of the high and low pools for each reporter/environment combination are reported in Supplementary Table 12. Cells were collected into sterile 1.5 mL polypropylene tubes with 1 mL of SC -his -lys medium and grown at 30 °C with rolling at least 26 h or until saturation. We added 1 mL of each culture to a 96-well plate with 600 µL of 40% glycerol and stored at −80 °C for subsequent genomic DNA extraction.

## DNA extraction and library preparation

We isolated genomic DNA from thawed glycerol stocks of the sorted segregant pools for whole-genome sequencing. We centrifuged 800 µL of each pool at 3,700 rpm for 10 min to pellet the cells and discarded the supernatant. To digest cell walls, we resuspended the cells in 800 µL of 1 M sorbitol, 0.1 M EDTA, 14.3 mM β-mercaptoethanol, and 500 U of Zymolyase lytic enzyme (United States Biological) and incubated for 2 h at 37 °C on a MixMate at 1,100 rpm. We re-pelleted the cells, removed the supernatant, and extracted DNA from the cells using the Quick-DNA 96 Plus kit (Zymo Research, Irvine, CA, USA), according to the manufacturer's instructions, including an overnight protein digestion in 20 mg/mL of proteinase K solution. We eluted the DNA using 35 µL of DNA Elution Buffer (10 mM Tris–HCl [pH 8.5], 0.1 mM EDTA) and determined DNA concentration on a Synergy H1 plate reader (BioTek Instruments, Winooski, VT, USA) in 96-well plates using the Qubit dsDNA BR assay kit (Thermo Fisher Scientific, Waltham, MA, USA).

We prepared the genomic DNA for short-read whole-genome sequencing on the Illumina NovaSeq platform using a previously established approach (Albert et al. 2014; Brion et al. 2020; Collins et al. 2022, 2023). We used the Nextera DNA library kit (Illumina, San Diego, CA, USA) in 96-well plates according to the

manufacturer's instructions, except that we used a 1:20 dilution of the Tagment DNA enzyme in Tagment DNA buffer. After library generation, we quantified the DNA concentration of each sample using the Qubit dsDNA BR assay kit (Thermo Fisher Scientific). For each 96-well plate, 10 µL of each sample was pooled and 1 mL of that pool was run in a large well on a 2% agarose gel. We extracted and purified the DNA in the 400 to 600 bp region using the Monarch Gel Extraction Kit (NEB) according to the manufacturer's instructions. The DNA from each of the gel extractions was quantified and pooled in equimolar amounts, and submitted for sequencing.

The University of Minnesota Genomics Center (UMGC) staff performed quality control assays as described in Collins et al. (2022, 2023) on the pooled library before sequencing. Briefly, library concentration was determined using PicoGreen dsDNA quantification reagent (Thermo Fisher Scientific), library size was determined using the Tapestation electrophoresis system (Agilent Technologies, Santa Clara, CA, USA), and library functionality was determined using the KAPA DNA Library Quantification kit (Roche, Penzberg, Germany). The submitted pooled library passed each quality control assay. The pooled library was sequenced on the Illumina NovaSeq with 150 bp paired-end reads. Of the samples used for analysis, the median number of reads produced per sample was 1,692,391, with the minimum being 304,256 reads and the maximum being 5,874,865 reads. UMGC performed sequence data de-multiplexing. Whole-genome sequencing data are available from the NIH Sequence Read Archive under Bioproject accession PRJNA1201919.

## Genetic mapping

QTLs were determined using an established approach for bulk segregant analysis (Michelmore et al. 1991; Ehrenreich et al. 2010; Albert et al. 2014; Brion et al. 2020). Code from Collins et al. (2022, 2023) was used to calculate allele frequencies via the following pipeline. From our whole-genome sequencing reads, we aligned reads to the *S. cerevisiae* reference genome (version sacCer3) using the BWA "mem" command (Li and Durbin 2009) and retained alignments with a mapping quality score above 30. Using samtools (Li et al. 2009), we retained uniquely aligned reads and removed PCR duplicates (command: "samtools markdup -S"). VCF files with allelic read counts at 18,871 high-confidence, reliable SNPs (Ehrenreich et al. 2010; Bloom et al. 2013) were produced using the command: samtools mpileup -vu -t INFO/AD -l.

We used adapted code from Collins et al. (2022, 2023) to calculate allele counts from the VCF files. Briefly, we excluded variants with allele frequencies lower than 0.1 or higher than 0.9 as in Albert et al. (2014) and Brion et al. (2020). We used MULTIPOOL (Edwards and Gifford 2012) to estimate logarithm of the odds (LOD) scores comparing a model in which the high and low degradation activity pools come from one population to a model in which these pools come from 2 different populations with different allele frequencies. As in Collins et al. (2022, 2023), we used the following MULTIPOOL settings: bp per centiMorgan = 2,200, bin size = 100 bp, effective pool size = 1,000. We called QTLs as loci with a LOD ≥ 4.5. Previous work has shown that this threshold produces a 0.5% false discovery rate for genetic mapping by bulk segregant analysis using TFT reporters (Collins et al. 2022). Confidence intervals (CIs) for each significant QTL were determined using MULTIPOOL and defined as a 2-LOD drop from the position in the QTL interval with the highest LOD score, which we defined as the QTL peak position. We calculated the RM allele frequency difference (ΔAF) between the high and low degradation activity pools using a smoothed allele frequency via a loess regression to account for random counting noise at individual sequence variants. In our scheme, a positive ΔAF indicates that the RM allele of a QTL is associated with higher UPS activity. The loess smoothed values were used for plotting and determining QTL effect sizes.

QTLs were called separately for each biological replicate. To determine QTLs that were detected in both biological replicates, we used a previously described approach (Collins et al. 2022, 2023). QTLs present in both replicates were defined as QTLs on the same chromosome with peaks within 100 kb and with the same effect direction (ΔAF sign). From our set of 694 QTLs across replicates, 416 QTLs (60%) were present in both replicates, with the remaining 278 QTLs detected in only 1 replicate. For QTLs present in both replicates, the left and right CI positions, peak position, LOD, and ΔAF were averaged and used for downstream analyses. For QTLs found in only 1 replicate, the left and right CI positions, peak position, LOD, and ΔAF were used without alteration.

The high and low populations of the second biological replicate of Phe in 4NQO did not have sufficient sequencing coverage to call QTLs. To replace these populations, we used additional populations we had collected during FACS of Phe in 4NQO from the first biological replicate. These additional populations consisted of cells with RFP/GFP ratios in the 3% to 5% area of the tails of the distribution. Therefore, both replicates used in data analysis of Phe in 4NQO came from the same original segregant pool.

## Comparison of QTLs from previous studies

QTLs for the Asn, Phe, Rpn4, and Thr reporters had been mapped previously in the same standard SC medium (Collins et al. 2022, 2023). For these 4 reporters, we analyzed 39 QTLs that were present in both replicates in SC in the present study and asked whether they were also present in at least one of the 2 replicates from the previous studies. If a QTL from the present study had a QTL whose peak was within 100 kb from Collins et al. (2022, 2023) in at least one replicate, we determined that this QTL was present in both studies. All QTLs present in both studies had the same sign of ΔAF. A 2-sample *t*-test was used to determine if there was a significant difference of LOD scores and absolute ΔAF between QTLs that were present in both studies compared with those found only in the present study.

## GxE in the QTLs

To determine GxE at individual QTLs, we compared loci between SC and each additional, distinct environment for each reporter. GxE at individual QTLs was classified as either (i) presence/absence or (ii) sign change. Presence/absence GxE QTL comparisons were defined as loci detected in both replicates of one environment, but where no QTL peak was found within 100 kb in either replicate of the other environment. Because a true QTL might be absent in our QTL set due to insufficient power, we only considered QTLs that were present in both replicates of one environment (and therefore are likely to be relatively strong in that environment) and absent in both replicates of the other environment.

Sign change GxE QTLs were defined as QTLs that were present in SC and a given environment but had ΔAF of a different sign. We considered QTLs to be present in both environments when their peak position occurred within 100 kb. We included QTLs in the sign change pairs that were found in both replicates of one environment and in one or both replicates of the other environment. The decision to include cases with a QTL in just one replicate of the other environment was made to increase our ability to detect sign changes, a type of GxE that turned out to be rare in our data. This decision comes at the potential cost of a slightly higher false discovery rate for sign change GxE. Of our 17 identified cases of

GxE, only 4 were based on just 1 QTL in one of the 2 environments, while 13 cases were based on seeing the given QTLs in all 4 applicable replicates. We did not observe cases in which the 2 replicates in the other condition both showed a QTL but with opposite sign within this condition.

If a pair of QTLs whose peaks were within 100 kb between SC and a given environment had the same sign of ΔAF, the pair was considered not to exhibit GxE. We included pairs of QTLs where a QTL was present in both replicates of one environment and present in either one or both replicates of the other environment.

## Results

### Experimental design overview

To study GxE in the UPS, we compared the UPS activity of 2 genetically divergent yeast strains toward 6 UPS substrates that engage multiple distinct UPS pathways in 8 environments including multiple starvation and chemical stressors, followed by genetic mapping of these UPS substrates in all 8 environments (Fig. 1a). We compared the BY laboratory strain, a close relative of the S288C reference strain, to RM, a vineyard isolate. These strains differ on average 1 nucleotide every 200 bp, providing abundant genetic variation that is known to affect molecular and cellular traits (Brem et al. 2002; Bloom et al. 2013; Albert et al. 2018; Brion et al. 2020; Nguyen Ba et al. 2022), including UPS activity (Collins et al. 2022, 2023).

In addition to a baseline medium, we selected 7 environments predicted to alter UPS activity. Throughout this paper, we consider SC medium ("SC"), a nutrient-rich medium, as the baseline environment for normal growth. UPS activity in SC was compared with 7 other environments intended to produce diverse impacts on cellular physiology, some of which have well-described effects on UPS activity (Supplementary Table 1). The environments included 3 "starvation" conditions predicted to decrease UPS activity: low glucose, low nitrogen, and a minimal medium formulation lacking amino acids (YNB). UPS protein degradation plays a critical role in the response to multiple chemical stressors. We therefore assayed 4 chemical stress conditions predicted to alter UPS activity by adding bortezomib (BTZ), L-azetidine-2-carboxylic acid (AZC), lithium acetate (LiAc), or 4-Nitroquinoline 1-oxide (4NQO) to SC. BTZ inhibits proteasomal protein degradation by tightly and selectively binding the 20S proteasome's catalytically active site, which causes proteolytic stress (Nunes and Annunziata 2017; Work and Brandman 2021). The proline analog AZC causes misfolding of nascent proteins (Rodgers and Shiozawa 2008; Work and Brandman 2021), resulting in proteotoxic stress that has been shown to increase UPS activity (Work and Brandman 2021). High salt concentrations cause the proteasome's 19S regulatory particle to dissociate from the 20S core particle (Glickman et al. 1998; Saeki et al. 2000). High salt has also been shown to inhibit

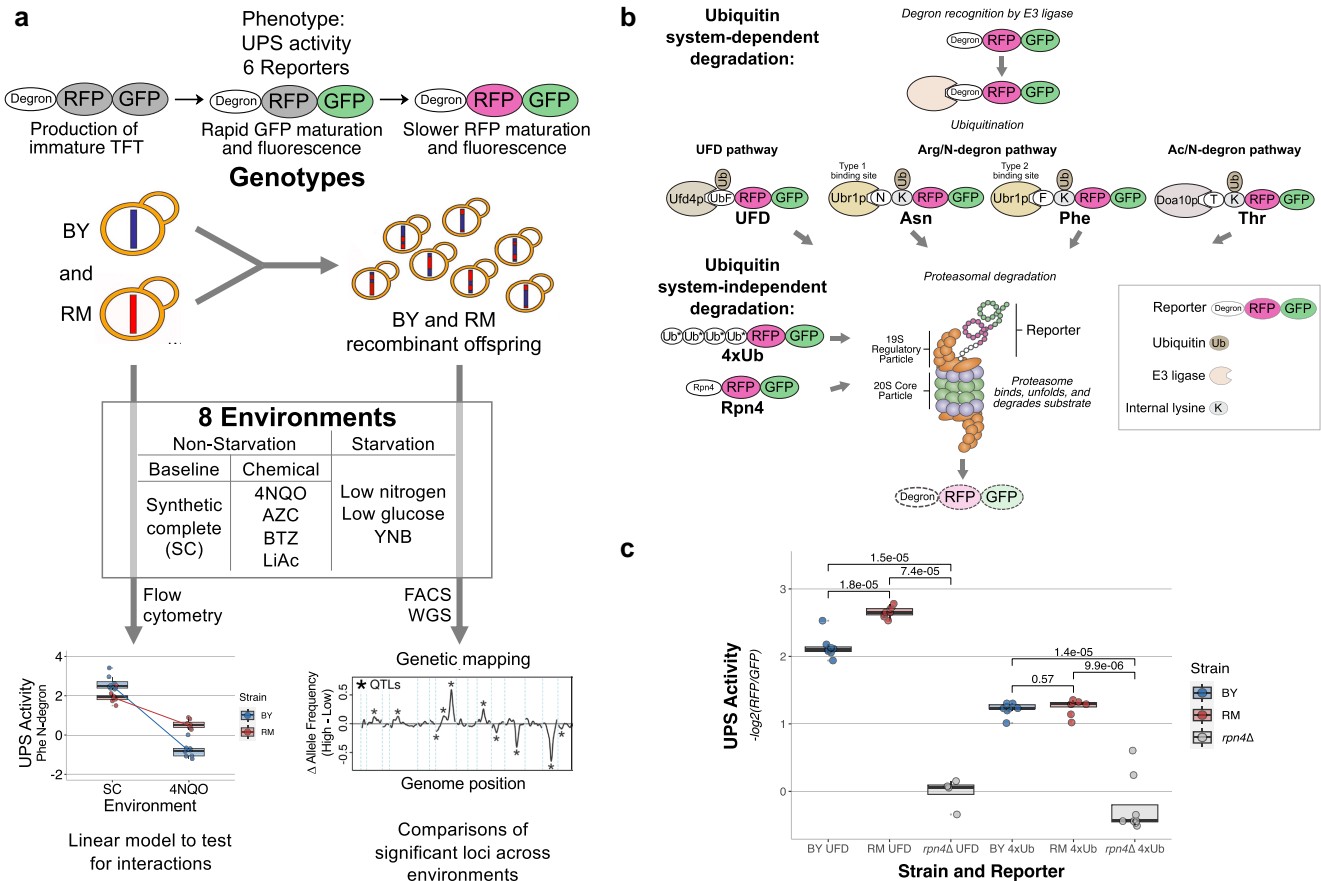

**Fig. 1.** Study design. a) Experimental design overview. TFT, tandem fluorescent protein timer; FACS, fluorescence-activated cell sorting; WGS, whole-genome sequencing; QTLs, quantitative trait loci. Genetic mapping trace from Collins et al. (2022). b) Simplified schematics of the 6 reporters used in this study to serve as substrates to measure ubiquitin system-dependent and -independent UPS pathway activity. UbF, ubiquitin with G76V substitution; Ub*, ubiquitin with G76V and K29/48/63R substitutions. Adapted from Collins et al. (2022, 2023). c) UPS activity from UFD and 4xUb reporters in BY, RM, and BY *rpn4*Δ. All samples had 8 replicates except *rpn4*Δ UFD (*n* = 4; see Methods). *P*-values from 2-tailed *t*-tests are indicated.

purified 20S core particles (Holtz et al. 2003). To create a high salt environment, we used LiAc, a chemical commonly used in yeast transformations. We sought to profile cells in an additional environment that may increase UPS activity. To this end, we used the mutagen 4NQO, which has been reported to increase global protein degradation (Burgis and Samson 2007).

To measure UPS activity, we used TFTs. TFTs are 2-color fluorescent protein constructs that provide high-throughput measurements of protein turnover in live cells (Khmelinskii et al. 2012; Khmelinskii and Knop 2014; Fig. 1b; Supplementary Table 2). The TFTs used here consist of a linear fusion of a faster-maturing GFP and a more slowly maturing RFP. Shortly after the TFT is produced (~5 min), the GFP folds and begins fluorescing, while the RFP matures and fluoresces over a longer time scale (>40 min). This difference in maturation kinetics can be used to measure the TFT's degradation rate. Specifically, if the TFT's degradation rate is faster than the RFP's maturation rate, then the $-\log_2$(RFP/GFP) ratio is directly proportional to the TFT's degradation rate (Khmelinskii et al. 2012; Khmelinskii and Knop 2014). Because the RFP and GFP are synthesized from the same mRNA transcript, the TFT ratio is independent of the expression level of the TFT (Khmelinskii et al. 2012; Khmelinskii and Knop 2014; Kats et al. 2018; Kong et al. 2021). TFTs can thus be used as reporters to measure protein degradation in genetically distinct cell populations where reporter expression may vary (Khmelinskii et al. 2012; Collins et al. 2022, 2023). Fusing a UPS degron to a TFT causes the construct to be rapidly targeted and degraded by the UPS. The $-\log_2$(RFP/GFP) output of a UPS degron-containing TFT thus provides a direct, quantitative measure of UPS activity that is sensitive to environmental and genetic perturbations that impact the UPS (Khmelinskii et al. 2012; Zhang et al. 2019; Collins et al. 2022, 2023). For example, deleting the proteasome transcription factor RPN4 causes a reduction in UPS activity that is reflected in a large and significant decrease in the $-\log_2$(RFP/GFP) of multiple TFTs containing UPS degrons (Khmelinskii et al. 2012; Collins et al. 2022, 2023).

We attached 6 degron-containing substrate sequences that engage multiple distinct UPS pathways to TFTs to create UPS activity reporters (Fig. 1b, Supplementary Table 2). The UPS reporters used here include 4 substrates targeted by the ubiquitin system ("ubiquitin system-dependent" reporters) and 2 substrates that are directly bound and degraded by the proteasome ("ubiquitin system-independent" reporters). The ubiquitin system-dependent reporters are sensitive to changes in any of the molecular events in UPS protein degradation, including substrate targeting by the ubiquitin system and proteasomal degradation. Three of the ubiquitin system-dependent reporters probe the 3 branches of the N-degron pathway, a UPS pathway in which a protein's N-terminal amino acid functions as a degron (Supplementary Table 2; Varshavsky 2011, 2019, 2024). These include the type-1 Arg/N-degron pathway (with the reporter containing asparagine as the N-terminal amino acid; "Asn"), which targets basic N-terminal amino acids; the type-2 Arg/N-degron pathway (phenylalanine; "Phe"), which targets bulky hydrophobic N-terminal amino acids; and the Ac/N-degron pathway (threonine; "Thr"), which targets acetylated uncharged N-terminal amino acids (Varshavsky 2011, 2019, 2024). N-degron pathways influence multiple aspects of cellular physiology by regulating protein abundance (Varshavsky 2011). To capture genetic influences on protein quality control-associated UPS activity, we constructed a TFT reporter that measures the activity of the ubiquitin fusion degradation pathway ("UFD")

(Johnson et al. 1995). In the UFD pathway, a noncleavable ubiquitin moiety acts as a degron that is recognized by the Ufd4p E3 ligase, which is involved in protein quality control through targeting misfolded proteins during conditions of proteotoxic stress (Johnson et al. 1995; Theodoraki et al. 2012; Devarajan et al. 2020).

To measure genetic effects on ubiquitin system-independent degradation pathways, we used 2 reporters containing degrons that are directly bound and degraded by the proteasome. The "Rpn4" reporter (Collins et al. 2023) contains the first 80 amino acids from the N-terminus of the Rpn4 protein, which are directly bound by the Rpn2p and Rpn5p receptors of the 19S regulatory particle of the proteasome (Xie and Varshavsky 2001; Ju and Xie 2004; Prakash et al. 2004; Ha et al. 2012). The second ubiquitin system-independent reporter contains a linear fusion of 4 ubiquitin molecules ("4xUb") (Stack et al. 2000), which functions as a degron that is recognized by the proteasome receptor Rpn13p (Thrower et al. 2000). The 4xUb degron thus provides a proteasome recognition element (a polyubiquitin chain) similar to that found on the vast majority of physiological UPS substrates, but without engaging the ubiquitin system (Thrower et al. 2000; Zhao and Ulrich 2010; Inobe et al. 2011; Martinez-Fonts et al. 2020). Because the Rpn4 and 4xUb degrons have different sizes, sequence compositions, and structures, we reasoned that they may be influenced by distinct sets of loci, as in our prior studies of ubiquitin-independent substrates (Collins et al. 2023). Our selection of substrates thus allowed us to capture genetic influences on the activity of multiple UPS pathways involved in physiological protein abundance regulation and protein quality control.

The UFD and 4xUb TFT reporters were developed for this study. We characterized these reporters using flow cytometry in BY, RM, and in a BY strain with reduced UPS activity due to the deletion of the RPN4 gene (hereafter, "rpn4Δ") (Xie and Varshavsky 2001). As expected, both reporters showed significantly lower UPS activity in the BY rpn4Δ strain than in the BY and RM strains when grown in SC (t-test, $P < 7.4\mathrm{e}{-5}$) (Fig. 1c). RM showed higher UPS activity than BY for UFD (t-test, $P = 1.8\mathrm{e}{-5}$), and there was no difference between BY and RM for 4xUb ($P = 0.57$). Thus, our 6 reporters provide high-throughput, quantitative, substrate-specific, in vivo measurements of UPS activity.

## Widespread GxE in UPS activity between 2 yeast strains

To estimate the extent of GxE in the UPS, we exposed BY and RM strains carrying one of the 6 reporters to the 8 environments and used flow cytometry to assay UPS activity in 20,000 cells in each of 8 biological replicates per combination of strain, reporter, and environment (Fig. 2a and b). Compared with the SC baseline, all 7 environments altered UPS activity in at least 1 strain and for at least 1 reporter (Fig. 2c and d, Supplementary Table 3). UPS activity significantly decreased in 39 of 80 comparisons and increased in 4 comparisons (t-test, Bonferroni-corrected $P < 0.05$). Of these increases, 3 were seen for BY in AZC (Asn, Rpn4, and 4xUb), in line with the reported modest increases in RPN4 expression caused by this treatment in a BY strain (Work and Brandman 2021). Notably, the effects of environment on UPS activity were highly strain-dependent. For example, all 4 cases of environmentally induced increases in UPS activity were seen in one but not the other strain (note the 4 significant points above zero in Fig. 2c and d). Together, these observations suggest widespread GxE in UPS activity.

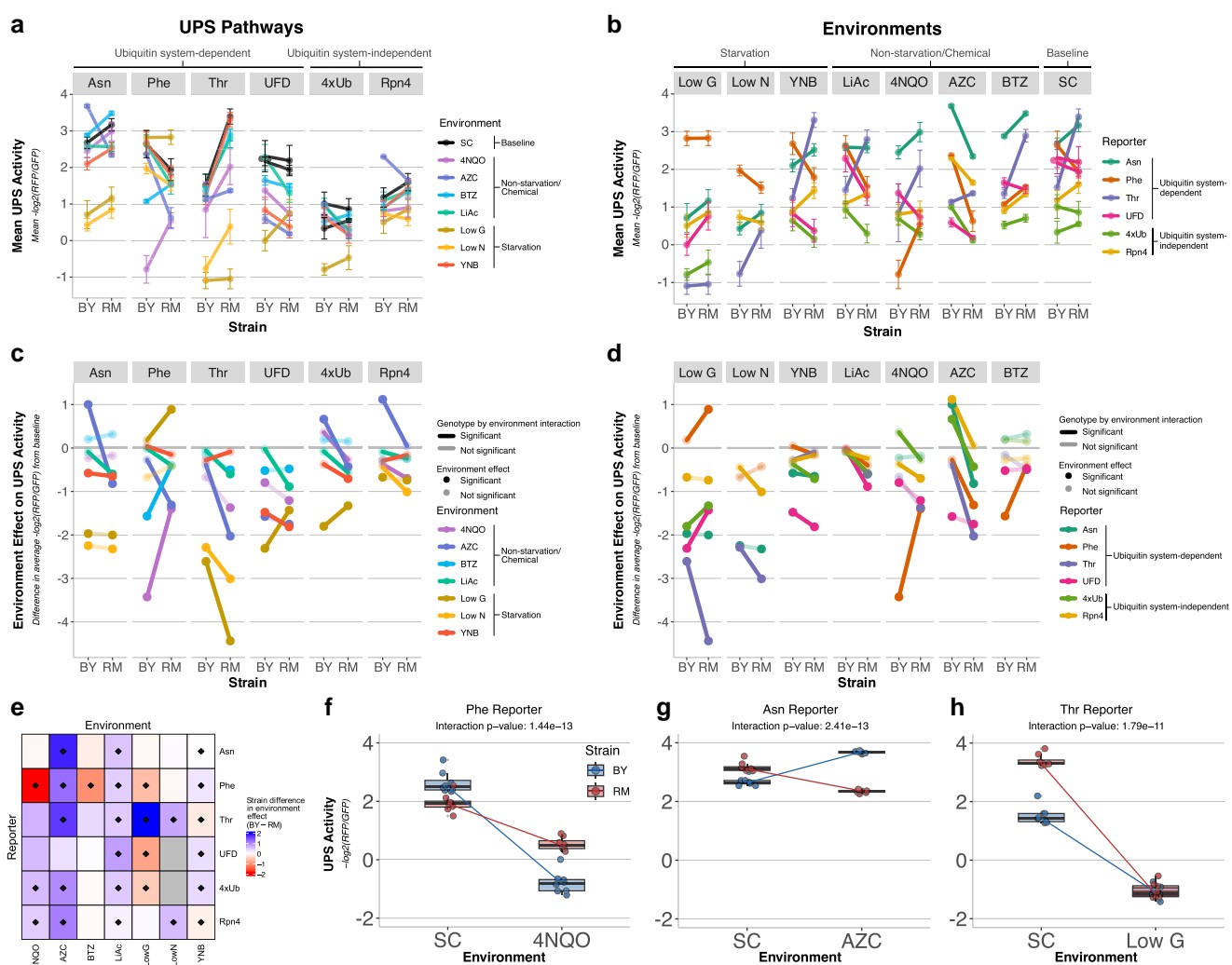

**Fig. 2.** GxE in BY and RM across reporters and environments. a and b) Mean UPS activity of 6 reporter and 8 environment combinations for BY and RM, based on 8 biological replicates. Error bars show standard deviations. c and d) Environment effects on UPS activity. Y-axis: the mean UPS activity among replicates in SC for the given strain was subtracted from that in the given environment to visualize environment effects. Negative values indicate that the environment caused a decrease in UPS activity compared with SC, and positive values indicate increased UPS activity. A value of zero means the environment did not affect UPS activity. Significant GxE terms (analysis of variance interaction term in the linear model, $P < 0.05$ after Bonferroni's correction) are highlighted by opaque lines. Opaque points indicate a significant difference ($t$-test, Bonferroni-corrected $P < 0.05$) in UPS activity between the given environment and SC for the specified strain. a and c) Data organized by reporter. b and d) Data as in a and c), respectively, but reorganized by environment. e) Heatmap summarizing strain differences in environment effect. Diamonds indicate significant GxE terms. a to e) All data shown in Supplementary File 2 and Table 3. Two sets of SC replicates were collected for UFD and 4xUb (see Methods—Flow cytometry). The UPS activity of the 4xUb and UFD reporters could not be reliably measured in low nitrogen, so no data were collected (see Methods). f to h) Reporter/environment combinations that exhibited the most significant GxE, ranked by $P$-value of the interaction term in the linear model. The center line of each box plot corresponds to the median of the 8 replicates, with the lower and upper hinges showing the first and third quartiles, respectively. Whiskers extend to 1.5 times the interquartile range and lines connect the respective BY and RM medians. d) Phenylalanine N-degron reporter in 4NQO. e) Asparagine N-degron reporter in AZC. f) Threonine N-degron reporter in low glucose.

To measure GxE more formally, we fit linear models that compared UPS activity for a given reporter between BY and RM and between SC and one of the other environments (Methods). GxE was detected in 27 (67.5%) of 40 tests (analysis of variance interaction term $P < 0.05$ after Bonferroni's correction), revealing numerous cases in which an environmental effect on UPS activity depended on the strain (Fig. 2e, Supplementary Table 3 and File 2). Specific examples and patterns among cases with significant GxE are presented below.

The most significant interaction effect ($P = 1e{-}13$) was seen for the phenylalanine N-degron reporter in 4NQO (Fig. 2f). 4NQO reduced the UPS activity measured by this reporter in both BY and RM (environment main effect: $P = 8e{-}19$; $t$-tests, $P \leq 1e{-}7$). However, the reduction was stronger in BY than in RM, such that BY had higher UPS activity than RM in SC, while it had lower activity than RM in 4NQO, resulting in a significant rank order change. The GxE term with the second most significant interaction effect was seen for the asparagine N-degron reporter in AZC ($P = 2e{-}13$) (Fig. 2g). In this case, AZC lowered UPS activity in RM but increased it in BY. This also resulted in a significant rank order change: in SC, RM had higher UPS activity than BY ($t$-test, $P = 3e{-}5$), while in AZC, RM had lower UPS activity ($P = 2e{-}17$). The GxE term with the third-smallest $P$-value was observed for the threonine N-degron reporter in low glucose ($P = 2e{-}11$)

(Fig. 2h). Here, RM showed higher UPS activity than BY in SC (*t*-test, $P = 4e-9$). However, in low glucose, UPS activity was much lower for both BY and RM, dropping to the 2 lowest (out of 96) mean UPS activity values in this experiment and removing the strain difference (*t*-test, $P = 0.72$) (Fig. 2a and b). These highly significant cases of GxE illustrate how the environment can differentially shape UPS activity in genetically different individuals.

Each reporter and environment showed at least one significant case of GxE (Fig. 2e). Among environments, LiAc and YNB showed GxE for all 6 reporters when compared with SC, while BTZ had the lowest number of significant interaction effects (1/6 reporters) (Fig. 2e). Across the dataset, a given environment decreased UPS activity in RM more than it did in BY in 20/27 cases. The 5 strongest (ranked by the absolute difference in strain response to the given environment) and most statistically significant cases of GxE were all for N-degron pathway reporters (Fig. 2c and e, Supplementary Table 3). N-degron pathway substrates require ubiquitin system targeting and, for 2 of the 3 measured substrates (Asn and Thr), preprocessing to produce functional N-degrons. The complex cascade of molecular events required to degrade these substrates may result in more frequent GxE in the genetics of UPS activity toward N-degrons relative to the other substrates tested here.

Some environments had consistent GxE effects across reporters (Fig. 2d and e). For example, in LiAc, GxE was seen for all reporters in that BY showed no significant change in UPS activity compared with SC for any reporter, while RM showed at least nominally significant reductions for all reporters (*t*-test for effect of environment per strain, $P \leq 0.04$). In AZC, BY had higher UPS activity than RM for all 5 reporters with significant GxE. This was either because AZC increased activity in BY while activity in RM was unchanged (Rpn4) or even reduced (Asn, 4xUb), or because RM experienced reductions in UPS activity, with no change in BY (Phe, Thr). Other environments had heterogeneous GxE effects across reporters. For example, glucose starvation decreased UPS activity in both strains but more so in RM than in BY for the Thr N-degron reporter; showed the opposite pattern with larger decreases in BY than in RM for UFD and 4xUb; and even increased degradation in RM with no change in BY for the Phe N-degron reporter (Fig. 2d). Our results reveal previously unappreciated complexities in the influence of strain background, substrate, and environment on UPS activity. Specifically, the effects of multiple environments commonly reported to consistently affect UPS activity were distinct, and in some cases discrepant, between strain backgrounds and UPS substrates.

## Genetic mapping of UPS activity

To identify genetic loci affecting UPS activity between BY and RM, we used a genetic mapping approach based on bulk segregant analysis (Michelmore et al. 1991; Ehrenreich et al. 2010; Albert et al. 2014; Brion et al. 2020; Collins et al. 2022, 2023; Methods). Briefly, UPS activity was measured in large, genetically diverse cell populations of haploid meiotic recombinant progeny ("segregants") generated by mating RM with BY strains harboring the UPS activity reporters. We exposed 2 independent segregant populations derived from independent BY/RM matings to each of the 8 environments and used FACS to collect pools of segregants from the extreme tails of the UPS activity distribution. Sorted segregant pools were then whole-genome sequenced to determine BY and RM allele frequencies. Genome regions where pools with high and low UPS activity differ significantly in allele frequency indicate quantitative trait loci (QTLs) that influence UPS activity (Fig. 1a, Methods). All QTLs included multiple genes

(Supplementary Table 4); likely candidate genes are presented throughout the text.

In the baseline SC condition, we identified 46 QTLs across the 6 UPS reporters. Four of these reporters (Rpn4, Asn, Phe, and Thr) were previously mapped in SC (Collins et al. 2022, 2023). To assess reproducibility, we compared QTLs identified here to those from our previous studies. We observed high concordance with prior results in terms of the QTLs detected, the corresponding allele frequency differences, and the overall shape of the QTL traces (Supplementary Fig. 1). Of the 39 QTLs identified here for these 4 reporters, 30 (77%) were also seen in Collins et al. (2022, 2023) (Supplementary Fig. 1e). The remaining 9 QTLs had significantly lower LOD scores and effects sizes, as measured by the absolute allele frequency difference (*t*-test, $P = 0.004$, and 0.0001, respectively; Supplementary Fig. 1f), suggesting that they may have been missed due to limited power. Similar results were obtained when comparing QTLs mapped earlier to those in our new data (24/31 QTLs replicated [77%], with reproducing QTLs having marginally larger LOD scores [$P = 0.07$] and larger effects [$P = 0.033$]). Thus, our approach represents a highly reproducible method for characterizing the genetics of UPS activity.

The UFD reporter (Fig. 1b), which was not mapped in our previous studies, had 5 QTLs in the baseline SC condition (Fig. 3, Supplementary Fig. 2a). These 5 QTLs include a QTL on chromosome XII (peak position at 950,450 bp) that was not seen for other reporters here or previously. This region contains the gene *RPN13*, which encodes a subunit of the 19S regulatory particle of the proteasome that acts as a ubiquitin receptor. There are multiple BY/RM promoter and missense variants at *RPN13*, along with a strong *cis*-eQTL for this gene (Albert et al. 2018), suggesting *RPN13* as a causal gene for this QTL.

The 4xUb reporter had 2 QTLs in the baseline SC condition, which is the fewest of all the reporters in SC (Fig. 3, Supplementary Fig. 2b). Both QTLs for 4xUb were previously identified for other reporters (Collins et al. 2022, 2023). Specifically, the QTL on chromosome VII contains *RPT6*, which encodes an ATPase of the proteasome's 19S regulatory particle. At a causal variant in the *RPT6* promoter, the derived RM allele broadly increases UPS activity toward multiple substrates of multiple UPS pathways by increasing *RPT6* expression (Collins et al. 2023), suggesting that this variant also affects 4xUb. The QTL on chromosome XV was previously seen for multiple reporters, including the ubiquitin system-independent Rpn4 reporter (Collins et al. 2022, 2023). The causal gene in this QTL is likely *IRA2*, a gene that underlies a *trans*-eQTL hotspot that affects the expression of thousands of genes and numerous growth traits (Smith and Kruglyak 2008; Lutz et al. 2022). While altered *RPT6* expression underlying the QTL on chromosome VII likely affects UPS activity directly, coding variants in *IRA2* (Lutz et al. 2022), which alter the activity of the Ira2p RAS signaling regulator, likely affect UPS activity indirectly. In sum, genetic mapping of UPS activity revealed complex, reproducible genetic architectures arising from novel and known loci.

## Heterogeneous genetic architectures shape UPS activity across pathways and environments

Across the 6 UPS reporters and 8 environments, we identified a total of 416 QTLs (Fig. 3a, Supplementary Fig. 3a and Table 4). All 47 assayed reporter/environment combinations (Methods) had at least 1 QTL present in both replicates (Supplementary File 3). The number of QTLs across environments and reporters ranged from 1 (4xUb in LiAc) to 19 (Asn in low glucose and Thr in low nitrogen) (Fig. 3b, Supplementary File 3). Among reporters, the largest number of QTLs was found for the Thr reporter ($n = 111$) and

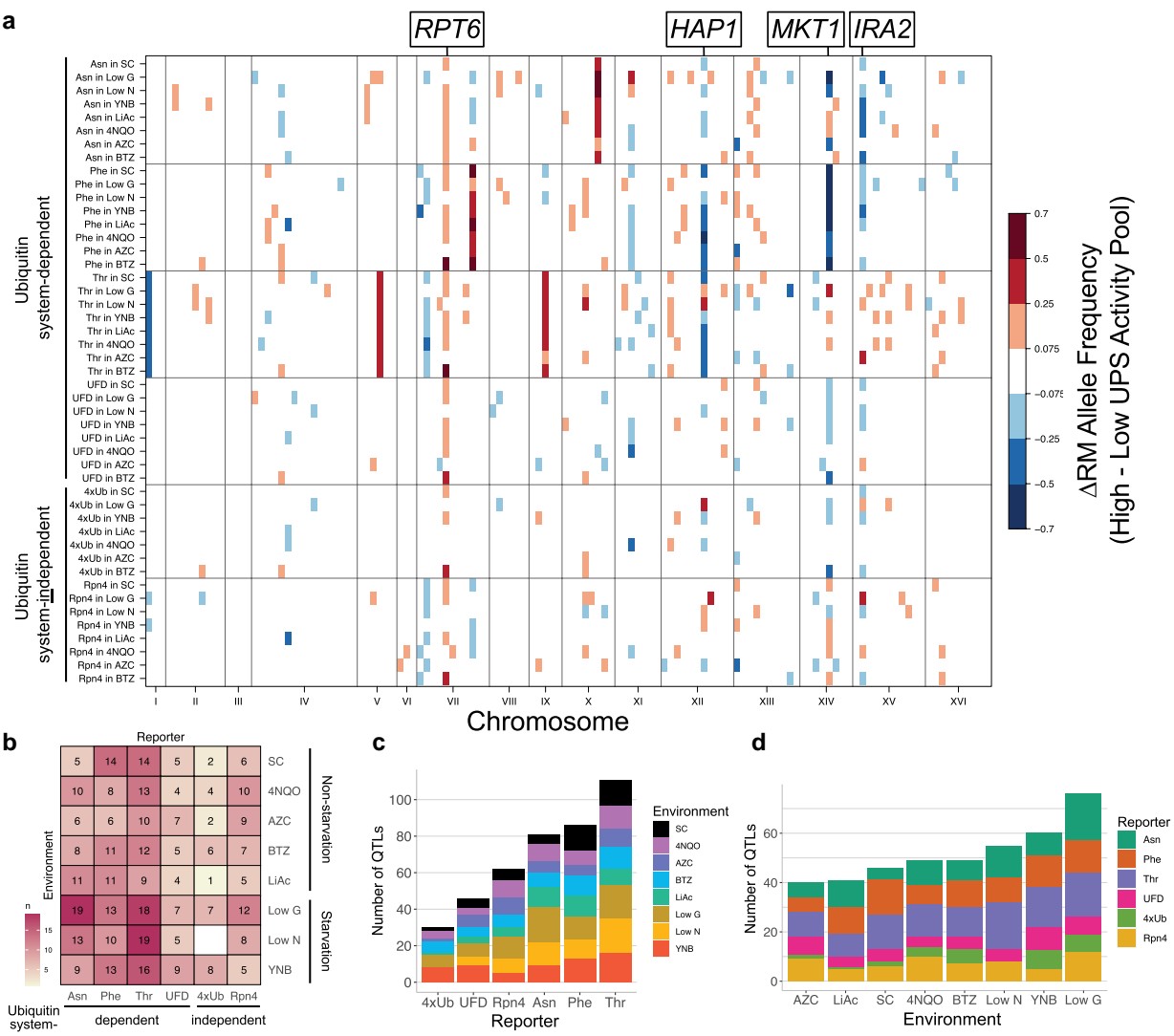

**Fig. 3.** UPS activity QTLs across reporters and environments. a) QTL mapping results for the 6 reporters across 8 environments. Colored blocks denote genomic bins that contain QTLs detected in each of 2 independent biological replicates, colored according to the direction and magnitude of the effect size, expressed as the RM allele frequency difference between high and low UPS activity pools. Genes in regions discussed in the text are indicated. b) Heatmap showing the number of QTLs per reporter/environment combination. The UPS activity of the 4xUb reporter could not be reliably measured in low nitrogen, so no data were collected (see Methods). c) Number of QTLs found per reporter and environment. d) Data as in c), but rearranged by environment.

the fewest for the 4xUb reporter ($n = 30$), when summing across all 8 environments (Fig. 3c). Thus, UPS activity is shaped by a wide array of heterogeneous genetic architectures across pathways and environments.

To examine these architectures in more detail, we collapsed combinations of reporters and environments into major UPS pathways and starvation vs. nonstarvation environments. Dividing the results by UPS pathways revealed that ubiquitin system-dependent pathways had significantly more QTLs (median = 10) than ubiquitin system-independent pathways (median = 6; Wilcoxon test, $P = 0.004$) (Fig. 3c), indicating greater genetic complexity. The substrates of the ubiquitin system-dependent pathways must undergo a complex cascade of molecular events that includes degron recognition, substrate ubiquitination, polyubiquitin chain extension, proteasome binding, and deubiquitination prior to degradation by the proteasome.

The genes encoding this machinery and *trans*-acting factors that influence their abundance and activity provide additional possibilities for causal genetic variation, likely contributing to the higher number of QTLs observed for these pathways compared with ubiquitin system-independent pathways. QTL patterns were unique for each substrate (see, for example, SC in Supplementary Fig. 3a), supporting our previous result that variation affects UPS activity in a substrate-specific manner (Collins et al. 2022, 2023).

Among environments and across all 6 reporters, the largest number of QTLs was found in low glucose ($n = 76$) and the fewest in AZC ($n = 40$) (Fig. 3d). The 3 starvation environments (low glucose, YNB, and low nitrogen) had significantly more QTLs (median = 10) than the nonstarvation environments (median = 7; Wilcoxon test, $P = 0.012$). One potential explanation for these results is that starvation environments may have more wide-reaching, systemic effects on cellular physiology than the chemical stressors tested

here. As a result, they may create more opportunities for variant effects that alter UPS activity through potentially highly indirect mechanisms. We also note that wild strains, such as RM, commonly undergo periods of nutrient deprivation, akin to the starvation environments tested here (Wenger et al. 2011; Hong and Gresham 2014). Consequently, RM may harbor alleles reflecting physiological adaptation to nutrient-poor environments, such as increased protein turnover for amino acid recycling (Vabulas and Hartl 2005). It is also possible that relaxed selection in the lab environment may have led to a loss of alleles associated with higher UPS activity in starvation conditions. Consistent with this notion, the RM allele increased UPS activity more often than the BY allele overall (binomial test, $P = 0.04$), and specifically for starvation environments (binomial test, $P = 0.04$) (Supplementary Fig. 3b and c), congruous with Collins et al. (2022, 2023). Together, our QTL mapping results show that in addition to being highly substrate-specific, genetic influences on UPS activity also depend on the environment.

## Loci with wide-reaching effects across pathways and environments

Many QTLs mapped to the same genomic locations (Fig. 3a). To quantify the number of unique locations, we counted the number of QTL peaks within 128 genomic bins of 100 thousand bp (kb) (Collins et al. 2022). Seventy-eight bins contained at least 1 QTL. The top 4 bins accounted for 28% (116/416) of the QTLs, illustrating that variation at a few locations underlies many of the genetic effects on UPS activity.

The bin on chromosome VII at 400 to 500 kb, which contains *RPT6,* harbored the most QTL peaks ($n = 32$, Fig. 3a). For these 32 QTLs, the RM allele always increased UPS activity. This is consistent with the previously characterized effects of variation at *RPT6* (Collins et al. 2023) and suggests *RPT6* is a causal gene in this bin. This bin contains other genes involved in the UPS with sequence variation between BY and RM: *SCL1*, which encodes the alpha 1 subunit of the 20S proteasome, and *RPN14*, an assembly-chaperone for the 19S regulatory particle. Thus, this locus likely shapes UPS activity directly via *RPT6* and potentially *SCL1* and *RPN14* as well.

Three bins on chromosomes XIV, XII, and XV contained 31, 26, and 26 QTLs, respectively (Fig. 3a). The direction of effect of the QTLs in these bins depended on the reporter and environment (Fig. 3a). These bins contain the genes *MKT1*, *HAP1*, and *IRA2*, respectively, which are all known to harbor variation that affects the expression of thousands of genes in *trans* (Albert et al. 2018). None of these genes has obvious connections to UPS function: *MKT1* encodes a poorly characterized protein that appears to bind certain mRNAs for genes with mitochondrial functions (Wickner 1987; Dimitrov et al. 2009); *HAP1* encodes a transcription factor that activates genes involved in osmotic stress (Gaisne et al. 1999); and *IRA2* encodes a regulator of Ras signaling (Tanaka et al. 1990). Therefore, the wide-reaching effects of these 3 genes likely alter UPS activity via indirect mechanisms.

## Few loci are specific to a single environment

Most QTLs were not specific to a single environment (Supplementary Fig. 3a), with 2 exceptions. First, at a locus on chromosome IV at 490 to 570 kb for LiAc (Supplementary Fig. 3a), the BY allele was associated with higher UPS activity than the RM allele for all 4 reporters with QTLs in LiAc in this region. This locus contains *ENA1*, *ENA2*, and *ENA5*, which all encode sodium pumps. *ENA1* and *ENA5* both contain multiple missense and frameshift variants between BY and RM, and the ENA locus

harbors structural variation among yeast strains (Warringer et al. 2011; Treusch et al. 2015). Collectively, this variation likely leads to the QTLs seen here in the LiAc environment, where higher activity of BY ENA alleles may reduce LiAc concentrations in the cell compared with RM alleles, alleviating osmotic stress on the UPS. Second, the *RPT6* locus on chromosome VII had QTLs in multiple environments; however, BTZ stood out in that all 6 reporters had QTLs with comparably large effects in this environment at this locus (Supplementary Fig. 3a). This locus also contains *PDR1*, which encodes a transcription factor that regulates genes involved in the yeast pleiotropic drug response (Moye-Rowley 2003), perhaps affecting cellular BTZ concentrations.

## Half of loci influencing the UPS show GxE

The distinct patterns of QTLs across pathways and environments show a high degree of GxE in UPS activity. To identify specific loci exhibiting GxE, we classified comparisons of QTLs into 3 categories based on pairwise comparisons between the baseline SC condition and individual environments: (i) presence/absence GxE, (ii) sign change GxE, and (iii) no GxE (Fig. 4a and b). While these comparisons do not account for GxE among non-SC environments, the patterns described here are broadly representative of other environment comparisons. To avoid confounding from substrate-specific variant effects, we exclusively compared QTLs from the same reporter. A total of 507 comparisons between QTLs in different environments were categorized according to this scheme (Fig. 4c and d, Supplementary Table 5). All reporters and all environments had loci exhibiting GxE, demonstrating that GxE is widespread in the genetics of UPS activity (Fig. 4c and d, Supplementary Figs. 4 and 5).

In "presence/absence" GxE, a QTL is detected in one environment in a comparison but not the other. We defined presence/absence GxE as instances where a QTL was detected in both replicates of one environment and where there were no QTLs within 100 kb in either replicate of the other environment (based on QTL peak positions as in prior studies, Collins et al. 2022, 2023; Fig. 4a and b, Methods). By requiring a locus to be present or absent in both biological replicates, we focused on the strongest cases of presence/absence GxE, reducing the chance that a locus is actually present in both environments but escaped detection in one environment due to insufficient statistical power. Presence/absence GxE was seen for half (254) of the 507 comparisons (Fig. 4c and d). For the Asn N-degron reporter, presence/absence GxE mostly involved QTLs that were absent in SC but present in the other environment (Fig. 4c). Conversely, for the Phe N-degron reporter, presence/absence GxE mostly involved QTLs that were present in SC but absent in the other environment (Fig. 4c).

In "sign change" GxE, a QTL is detected in both environments in a comparison but with opposing effects between environments. We formally defined sign change GxE as comparisons of QTLs with opposite effect direction that were detected in both replicates of one environment and at least one replicate of the other environment (Fig. 4a and b, Methods). Sign change GxE was much rarer than presence/absence GxE and was observed at 17/507 comparisons (Fig. 4c and d). Of these, 13 cases showed the sign change in both replicates of both of the given conditions, and in the remaining 4 cases only one replicate showed the sign change, while the other replicate showed no QTL. Sign change GxE was seen for all reporters (Fig. 4c) and in the low nitrogen, low glucose, and AZC environments (Fig. 4d).

A third form of GxE (magnitude GxE) that occurs when a locus has the same direction but differing magnitudes of effect cannot be reliably detected with our experimental design. Therefore,

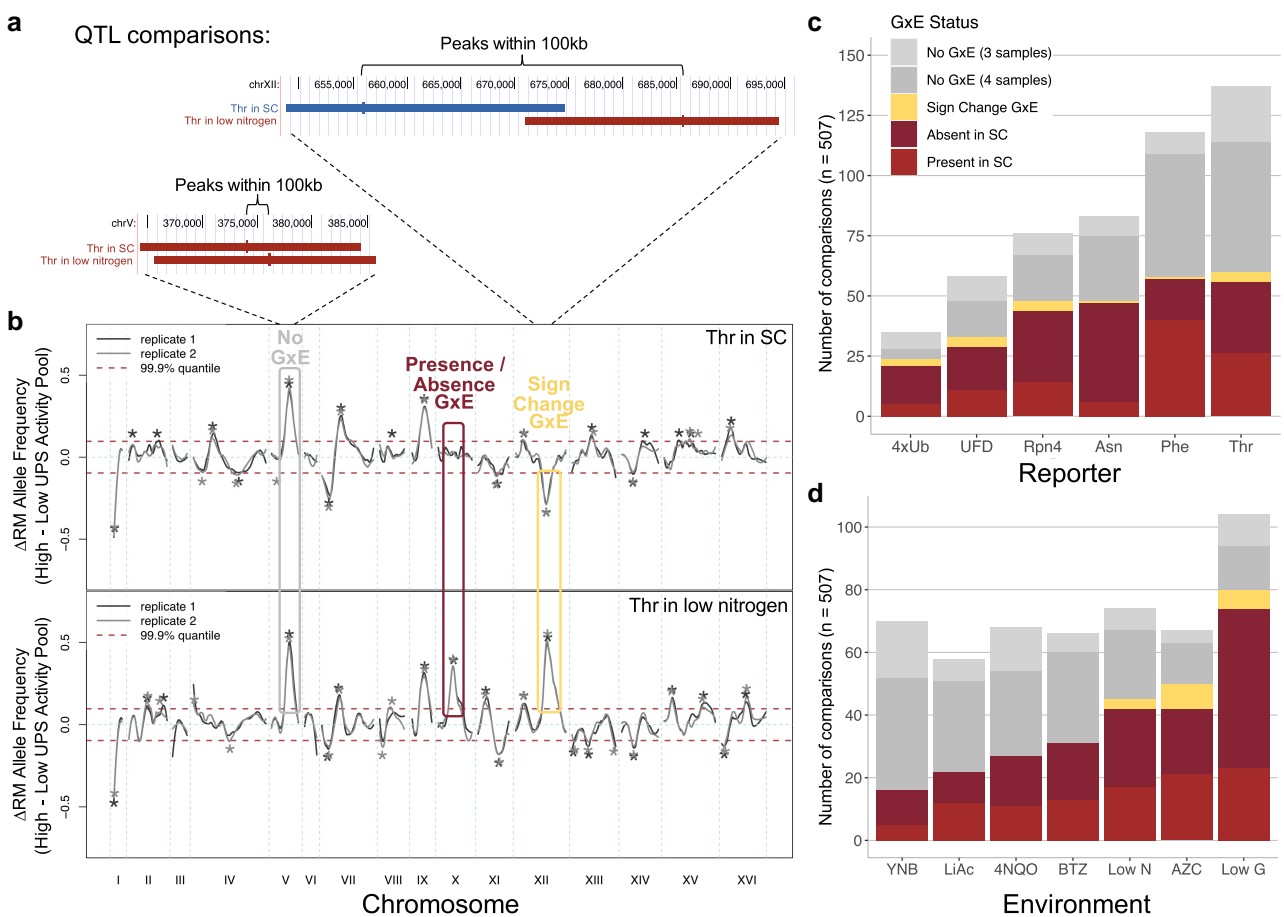

**Fig. 4.** GxE at individual loci. a and b) Examples of QTL comparisons that show presence/absence GxE and sign change GxE. a) Peaks (short vertical lines) of QTLs (horizontal bars; positions averaged across the 2 biological replicates) in the 2 compared environments must be within 100 kb to be considered present in both environments. Color indicates direction of allelic effect as in Fig. 3a. Blue: BY allele increases UPS activity; red: RM allele increases UPS activity. Images generated using UCSC Genome Browser (Nassar et al. 2023). b) QTL traces for the Thr N-degron reporter in SC (top) and in low nitrogen (bottom). Boxes highlight examples of the 3 categories of pairwise comparisons of loci. c) Pairwise comparisons of loci between SC and other environments, across reporters. Light gray indicates comparisons where a QTL was present in both replicates of one environment and only one replicate of the other environment, with the same direction of effect. d) Data as in c), but rearranged by environment.

QTLs within 100 kb with the same effect direction between environments are considered not to exhibit GxE (Fig. 4b).

Across presence/absence and sign change GxE, ubiquitin system-independent pathways had a slightly larger proportion of QTLs with GxE than ubiquitin system-dependent pathways (Wilcoxon test, $P = 0.046$) (Supplementary Fig. 4). There was no difference in the proportion of loci with GxE between starvation and nonstarvation environments (Wilcoxon test, $P = 0.82$) (Supplementary Fig. 4), as illustrated by the fact that both the fewest (YNB), and the most (low glucose) QTLs exhibiting GxE were seen for starvation environments (Fig. 4d). These results show that half of the loci we detected that shape UPS activity are subject to GxE, primarily via presence/absence of a given locus in different environments.

## GxE arises from loci that shape the UPS through direct and indirect mechanisms

We examined the location of QTLs exhibiting GxE using the binning strategy described above, and observed a nonuniform distribution (Fig. 5a). Almost all (71/78, 91%) bins that contained any QTLs contained loci that exhibited GxE (Fig. 5b). The 6 bins with the most GxE cases contained 27% (73/271) of all GxE cases

(Fig. 5a). We searched these top 6 bins for candidate genes and noticed that 2 bins contained candidate genes clearly related to the UPS. A bin containing *RPT6* had 10 GxE cases. All of these were presence/absence GxE, such that whenever the QTL was present, the RM allele increased UPS activity (see also Fig. 3a). Thus, variation at *RPT6* either has an effect in a given condition or it does not, but the RM allele (which increases Rpt6p abundance) does not reduce UPS activity in the conditions assayed here. In contrast, the remaining top bins with candidate genes each contained cases of presence/absence as well as of sign change GxE. A bin at 0 to 100 kb on chromosome XIII had 14 cases of GxE (Fig. 5a). This bin contained *BUL2*, which encodes a substrate adaptor for the Rsp5p E3-ubiquitin ligase complex. Bul2p regulates the cellular response to multiple stressors, and BY/RM variants in *BUL2* influence multiple cellular and organismal traits, including chronological aging and telomere maintenance (Kwan et al. 2011). Two of the top GxE bins corresponded to the *trans*-eQTL hotspots at *IRA2* (15 GxE cases) and *HAP1* (11 GxE cases), respectively (Fig. 5a). The 2 remaining top bins (chromosome IV at 400 to 500 kb and chromosome XIII at 300 to 400 kb) did not contain obvious candidate genes. Overall, QTLs with cases of GxE tended to be located in bins that also contained a large number of QTLs of

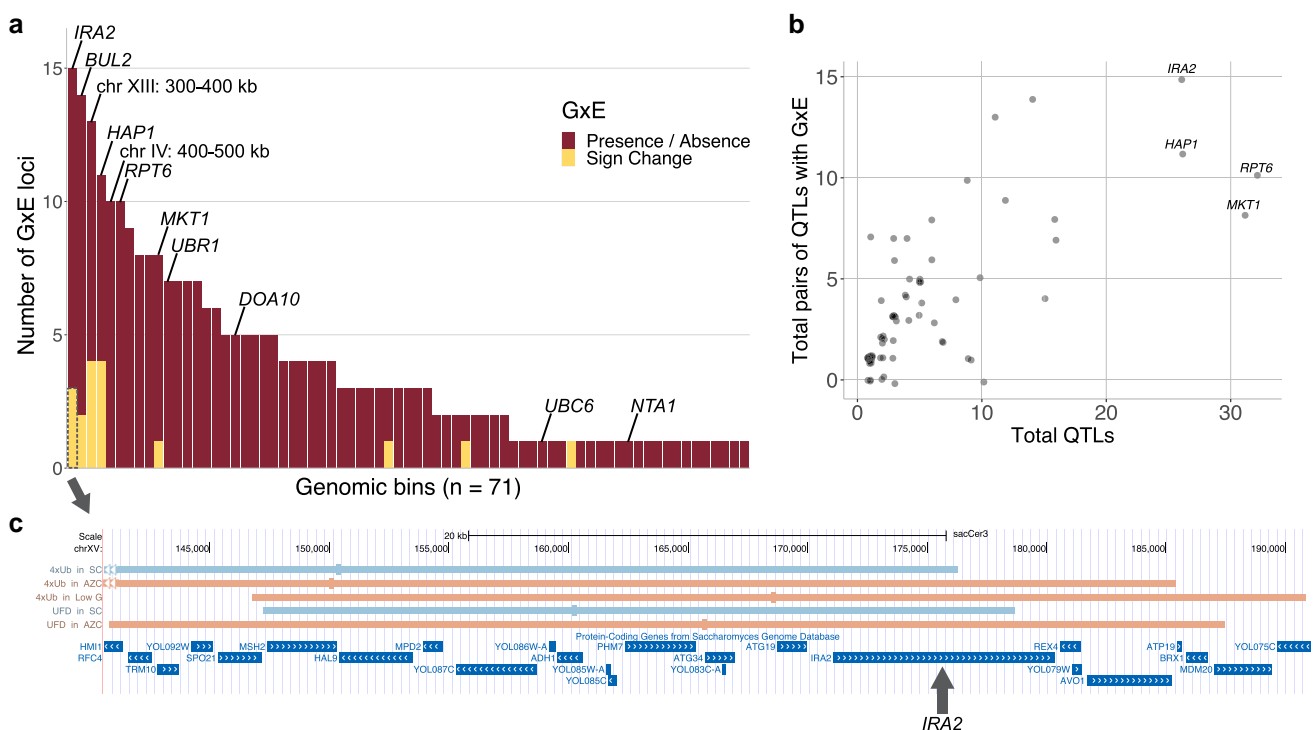

**Fig. 5.** Patterns of QTLs with GxE across the genome. a) Distribution of QTL comparisons that exhibited GxE in 100 kb bins. Shown are the 71 bins that contained QTL comparisons with GxE out of all 128 genomic bins. Bins are sorted based on the number of GxE cases they contain, followed by genomic position to break ties. Genes in bins discussed in the text are indicated. b) Correlation of the number of total QTLs and of QTL comparisons exhibiting GxE for each of 78 genomic bins that contained at least 1 QTL. Spearman's correlation, $\rho = 0.73$, $P = 5e-14$. Bins with the most QTLs are indicated. c) Locus plot showing 5 QTLs involved in 3 sign change pairs (1 for UFD, 2 for 4xUb) in the bin with the most cases of GxE. Genes in this region are indicated, with *IRA2* highlighted by the arrow. QTL CIs are shown as horizontal bars and peaks are indicated by the small rectangle within each QTL. QTL colors indicate the direction and strength of effect as in Fig. 3a. The QTLs for 4xUb in SC and AZC extend leftwards to position 124,750 and 112,300, respectively. Image generated using UCSC Genome Browser (Nassar et al. 2023).

any type (Spearman's correlation across 78 bins with any QTLs, $\rho = 0.73$, $P = 5e-14$) (Fig. 5b), suggesting that loci with wide-reaching effects are also generally prone to GxE. Collectively, these data show that a small portion of the genome harbors much of the GxE seen at individual QTLs. Loci subject to GxE occur throughout the genome but are clustered in regions with many QTLs, including those caused by variation in core UPS genes as well as indirect, pleiotropic regulators.

Sign change is a particularly interesting form of GxE, in which a locus results in opposite effects on UPS activity depending on the environment. There were 17 cases of QTL comparisons with sign change GxE. These 17 cases included 5 instances where QTLs with effect directions opposite that in SC were seen in multiple non-SC environments (Supplementary Table 6) and therefore consist of only 29 unique QTLs instead of 34 (Fig. 5c, Supplementary Fig. 5 and Table 7). Examination of the peaks and CIs of the QTLs involved in sign changes revealed that they clustered at 7 genomic regions (Fig. 5c, Supplementary Fig. 5a to f). Of these regions, 3 corresponded to the *trans*-eQTL hotspots at *IRA2* (5 QTLs involved in sign changes, Fig. 5c), *HAP1* (7 QTLs, Supplementary Fig. 5a), and *MKT1* (3 QTLs, Supplementary Fig. 5b). These genes have no known connections to the UPS. Two additional regions contained no genes with known UPS functions: a region from 280 to 470 kb on chromosome XIII with 6 sign change QTLs (Supplementary Fig. 5c), and a sign change pair (for Rpn4 in AZC), which was located ~98 kb from *UBC6* (Supplementary Fig. 5d). *UBC6* encodes the E2 ubiquitin-conjugating enzyme of the Ac/N-degron pathway (Varshavsky 2024) and Collins et al. (2022)

identified *UBC6* as a causal gene affecting degradation of the Thr N-degron reporter. However, its function in the ubiquitin system and relatively large distance from the sign change pair makes it unclear if *UBC6* is a causal gene for the ubiquitin system-independent Rpn4 reporter, with no other obvious candidate genes in this region. The remaining 2 regions with sign change QTLs contained likely causal genes with direct UPS functions: 1 sign change pair (for UFD in AZC) at *RPT6* (Supplementary Fig. 5e), and 4 such QTLs at *BUL2* (Supplementary Fig. 5f). Notably, none of the remaining causal genes that we previously determined to shape UPS activity (*UBR1*, *UBC6*, *NTA1*, and *DOA10*; Collins et al. 2022) had sign changes between SC and other environments, even though they did show presence/absence GxE. All of these genes encode core UPS components. Thus, loci with genes that may affect UPS activity in a direct fashion (*RPT6* and *BUL2*) accounted for only 21% (6/29) of the QTLs involved in sign change GxE. Most of the sign change QTLs (52%, 15/29) appear to arise from genes that shape UPS activity indirectly. In sum, these results suggest that GxE in the UPS, especially sign change GxE, is mostly caused by indirect mechanisms, such as widespread changes in gene expression due to *trans*-eQTL hotspots, rather than by variation in core genes directly involved in the UPS.

## Discussion

To characterize GxE in the genetics of protein degradation, we measured UPS activity toward 6 distinct substrates in single cells of 2 strains of *S. cerevisiae* and their progeny across 8

environments. GxE was pervasive between the 2 strains. The activity of every measured pathway was modified by the environment, and all of the tested environments led to GxE.

The BY and RM strains differed greatly in how they responded to a given environment. Remarkably, for some combinations of reporter and environment, UPS activity increased in one strain but decreased in the other. Previous studies of the UPS using some of the environments studied here were based on lab strains, suggesting that some published treatment effects are not broadly generalizable to the *S. cerevisiae* species as a whole. Strain-dependency of treatment effects has been widely documented, including between closely related strains of mice (Simon et al. 2013) and yeast (Matheson et al. 2017; Elserafy and El-Khamisy 2018), and our results reinforce the value of studying physiological effects in multiple genetic backgrounds including those that have evolved in different environments.

The interaction of genetics and environment was apparent for loci shaping UPS activity. All reporter/environment combinations had unique QTL patterns, and about half of the QTL comparisons we conducted revealed evidence of GxE. The presence of a given QTL in one but not another environment was by far the predominant form of GxE, making up 94% of the detected GxE cases. In spite of the large number of QTLs showing GxE, no QTLs were entirely specific to a particular environment, with the ENA locus in LiAc as the closest exception. Most QTLs were seen in several, but not all environments. Thus, the distinct QTL patterns for specific pathway/environment combinations were formed from subsets of the total set of QTLs identified across the entire study.

The QTLs we identified for different environments and pathways, including QTLs with GxE, tended to be clustered at certain genomic locations, as seen in our previous work on the UPS (Collins et al. 2022, 2023) and reflecting work on GxE in yeast gene expression (Smith and Kruglyak 2008). The number of GxE cases in a region was strongly correlated with the overall number of UPS activity QTLs in that region (Fig. 5b), suggesting that rather than only occurring at a small set of specific environmentally sensitive loci, GxE commonly occurs whenever genetic variation affects UPS activity.

A genome region with numerous QTLs and cases of GxE contained *RPT6*, which we earlier showed to contain a causal promoter variant that influences UPS activity toward the Rpn4 degron and substrates of the Arg/N-degron and Ac/N-degron pathways (Collins et al. 2022, 2023). Here, the *RPT6* locus affected all 6 assayed reporters in at least 1 environment. Thus, the causal variant in the *RPT6* promoter appears to have wide-reaching effects on UPS activity. *RPT6* can be considered a "core" gene under an omnigenic model of complex traits (Boyle et al. 2017), in which core genes encode proteins that are directly related to the given trait, while "peripheral" genes act as indirect regulators that shape complex traits through *trans*-acting effects on core genes. Here, we detected QTLs and presence/absence GxE at all core UPS genes we previously showed to cause UPS activity variation (*RPT6*, *UBR1*, *UBC6*, *NTA1*, and *DOA10*).

Many individual QTLs, QTLs with GxE, and in particular sign change GxE, occurred at genes known to cause *trans*-eQTL hotspots, where variation at a single gene affects the expression of hundreds or thousands of genes throughout the genome (Brem et al. 2002; Smith and Kruglyak 2008; Zhu et al. 2008; Albert et al. 2018). These loci exert broad effects through indirect mechanisms in which they alter cellular states that in turn affect many downstream traits, including growth in diverse conditions (Bloom et al. 2013; Renganaath and Albert 2025). Their many pleiotropic effects include UPS activity (Collins et al. 2022), for

which they can be interpreted as "peripheral" genes under the omnigenic model. Our results show that these hotspots are also focal points for GxE in UPS activity. Their indirect mode of action likely creates numerous opportunities for variant effects on physiological or molecular processes, which are not directly involved in UPS protein degradation but nonetheless affect UPS activity in an environment-dependent manner. Given the widespread effect of *trans*-eQTL hotspots on gene expression, it is likewise possible that GxE emerges through hotspot effects on the expression of distinct sets of genes in various environments. This stands in contrast to variation at core UPS genes, which was less prone to sign change GxE. The direct effects of core genes on the UPS may be less responsive to environmental influences compared with the more indirect, pleiotropic hotspot modulators.

Our study had several limitations. Because the bulk segregant mapping design we employed makes it difficult to rigorously detect differences in magnitude of locus effect in the same direction, we did not search for such loci even though magnitude GxE could be prevalent (Smith and Kruglyak 2008; Cubillos et al. 2014). As such, our GxE QTL results are conservative in that they only search for extreme cases in which a QTL is present in only 1 environment or switches sign. Due to linkage in the segregant population, a sign change QTL could reflect 2 presence/absence loci in close proximity. Only experimental determination of causal genes can ultimately rule out this possibility, although we note that GxE in gene expression has previously been shown to arise from variation in the single *IRA2* gene (Smith and Kruglyak 2008). Future work could map the underlying causal genes and variants in the loci identified here to clarify how they cause GxE.

Our results show that the genetic architecture of UPS activity is complex, substrate-specific, and subject to extensive GxE. Different UPS pathways affect the degradation of distinct substrates, different environments challenge proteostasis in different ways, and genetic variants differ in how they affect a given pathway in a specific environment. This complexity underscores the challenge of predicting molecular and organismal phenotypes from genotype. At the same time, our finding that hotspots of transcriptional variation found in previous studies overlap with those that shape posttranslational processes represented by UPS activity suggests that there is a shared, tractable set of causal genes and variants whose identification may enable prediction of the effects of genetic variation across the gene regulatory cascade. Our results are an important step forward in understanding how genetic variation and environmental factors modulate an essential molecular process, UPS protein degradation, that is relevant to numerous organismal traits across eukaryotic life. Characterizing the mechanistic link between genetic effects on UPS activity and organismal traits is likely to expand our understanding of how genetic influences on molecular processes give rise to variation in complex traits, including health and disease.

## Data availability

All Supplementary tables are included in Supplementary File 1. Code used to analyze data and generate plots is available at https://github.com/randiraphd/gxe_ups_activity. Flow cytometry data are available in Figshare at https://doi.org/10.6084/m9.figshare.c.7976957. Whole-genome sequencing data are available through the NIH Sequence Read Archive under Bioproject accession PRJNA1201919.

Supplemental material available at GENETICS online.

## Acknowledgments

We thank members of the Albert laboratory for experimental guidance and feedback on the manuscript. The authors acknowledge the University of Minnesota Genomics Center (UMGC) for providing resources that contributed to the research results reported in this paper. We thank Rashi Arora and the University of Minnesota Flow Cytometry Resource staff for technical assistance with flow cytometry and FACS.

## Funding

This work was supported by the National Institutes of Health grant R35GM124676 to FWA.

Conflicts of interest: None declared.

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

*Editor: P. Wittkopp*