## [Peer Review File · Genetics]

Genotype-by-environment interactions shape ubiquitin-proteasome system activity

Randi Avery, Mahlon Collins, and Frank Albert

NOTE: The reviews and decision letters are unedited and appear as submitted by the reviewers.

In extremely rare instances and as determined by a Senior Editor or the EIC, portions of a review may be redacted. If a review is signed, the reviewer has agreed to no longer remain anonymous.

The review history appears in chronological order.

Review Timeline:

Submission Date:	2024-12-25
Editorial Decision:	2025-02-09
Resubmission Received:	2025-06-18
Accepted:	2025-07-21

Summary :

The authors measure the activity of the yeast ubiquitin-proteasome system (UPS) in seven environments, using six fluorescent reporters – two of which have been newly designed for this study. Most of these environments are stresses which are known to affect UPS activity in defined ways, yet many discordant effects are observed between strains BY and RM. This thus reveals genotype-by-environment (GxE) interactions for UPS activity. These interactions are next investigated through QTL mapping, which shows that variation is mostly found at a limited number of loci. Some contain genes which are directly involved in the UPS, while others interestingly contain genes known as *trans* regulatory hotspots, and even genes involved in response to some of the stressors. More importantly, the authors investigate how GxE arise from these QTLs, which further emphasizes that variation among strains and environments is caused by a small number of loci, and that not all are directly related to the UPS. This work reveals and underappreciated potential for GxE at the level of protein degradation, while also highlighting how these interactions may vary across UPS pathways. This has many interesting implications, for instance regarding the evolution of protein abundance.

Strengths :

- The experimental approach developed by this group and used in the current study is innovative and particularly interesting. Applying it to GxE seems like a logical and important next step after their previous papers. The tandem fluorescent protein timers used should especially be more widely known.
- The work is very rigorous, with biological replicates throughout and the use of a common starter culture to inoculate the different environments
- The dissection of GxE into three types at the level of individual loci provides valuable insights into the different mechanisms by which variation in protein degradation can arise. Some additional analyses could however potentially help in making them clearer for readers (see below).

Major comments :

- I have not been able to access the code related to this work. The Github link in the *Data availability* section leads to an empty repository. This is certainly not intentional from the authors, but it has nonetheless limited my understanding of the analyses which have been performed.
- My main concern about the paper is that it may be slightly too descriptive. I found it difficult to summarize the findings into a few “take-home” messages, which also made it harder to follow the reasoning from one section to the next. It

is however entirely clear to me that the authors did very interesting and relevant work. This data seems to provide lots of insights into how variation in protein degradation can emerge, but they are not always made clear to the readers. I would especially be interested in more comparative analyses of the properties of loci which give rise to QTLs with and without GxE. I'm under the impression that adding some more summary figures and performing additional such analyses would make the paper more approachable for readers. I hope that my suggestions will prove helpful, but I am aware that the authors are in a much better position than me to assess their relevance.

- In general, it seems to me that the fact that there are both ubiquitin-dependent and ubiquitin-independent reporters, which give complementary information on the UPS, was not emphasized enough throughout the results. Since so much data is shown, it's easy for the reader to forget this information (and its significance), especially as not all figure panels make it clear.
- Figure 2 could probably benefit from one (or two) figure(s) summarizing the frequency and direction of effects, across reporters and environments. I found panels A and B slightly difficult to interpret, and replacing one of them with a figure showing mean effects and occurrences of positive and negative changes might make it easier for readers to notice the trends in the data. Similarly, Supp Fig 1B-C show how GxE frequency varies between reporters and environments, which seems important enough to be included in a main figure.
- Figure 3, and especially panel A, is striking, but I wonder if it could benefit from a panel showing how the number of QTLs detected is related to the magnitude of environmental effects seen in Fig 2. I'm under the impression that this would help connect these two sections of the manuscript. Similarly, are there more or less QTLs when a GxE is observed between BY and RM on Fig 2?
- Fig 5A is especially interesting, but I am unsure about the relevance of the corresponding panel B. Since the pairs showing GxE are a subset of the total pairs, and I don't understand how this correlation is informative. Perhaps, the saturation effect which is visible on the scatterplot is what should be emphasized? Otherwise, it might be worthwhile to replace this panel with another one showing the percentage of QTL bins showing GxE which also contain UPS-associated genes. This analysis has been performed for sign change GxE (lines 477-480), but I wonder why it is not also discussed for "presence/absence" and "no GxE" cases.

I may be overstepping my role as a reviewer with the suggestions above, and I apologize if that is the case. My general impression when reading the manuscript was that the figures may be too focused on showing the data itself instead of

what should be concluded from it. This made it harder for me to understand and follow the paper, but I might be part of a very small minority of readers in that regard.

- An additional concern I have is about the replicability of previously detected QTLs (lines 292-301). While I trust the authors, I find that this part of the paper could be more convincing. My understanding is that only the QTLs which have been identified in the present study but not reported in the two previous studies are discussed. I suppose that any QTLs which were detected in the previous studies but not replicated in this one also have lower LOD scores and effect sizes, but I think it would be important to mention it as well.

Minor comments :

- The introduction nicely exposes both the UPS system and the variability of protein degradation. I however wonder whether the paragraphs discussing GxE (lines 31-69) could benefit from being shortened a little. It seems to me that some of these examples distract a bit from the main point of the paper. This is however highly subjective and I may be completely wrong in this regard.
- In general, I think there may be a bit too much back-and-forth between figures and panels throughout the results section, which makes it harder to follow:
 - Fig 1A is first referenced at line 139, but it is not fully explained until line 291. This confused me at first, although the detailed explanation of the experimental approach at lines 281-291 was crystal clear.
 - In the next section (lines 222-275), Fig 2A-C is first referenced (lines 232, 238). Then, Fig 2D (line 240, 2E (line 244) and 2F (line 248) are discussed in order. After this however, the text goes back to Figs 2A, B & C, which are referenced multiple times (lines 252, 255, 258, 263 and 275). A way to limit this back-and-forth could be to first fully describe the environmental effects on UPS activity, which is currently split between lines 225-233 and lines 263-268, before describing the significant GxE detected.
 - In addition, although this is not related to the organization of the results section, one thing I found slightly confusing about Fig 2A-B is that I was not totally sure whether the baseline was the same for BY and RM, or rather strain-specific.
 - The same kind of back-and-forth between panels occurs for Fig 3. The first panel to be referenced individually is 3D (line 308), later followed by 3B (lines 309, 313, 335), 3C (line 344) and then 3A (line 357).

- I may be mistaken, but it seems that the 4NQO abbreviation is introduced at line 154 without being first defined.
- As a final (very) minor comment, there seems to be a mistake in the legend of Supplementary Fig 1C. It should probably read “Data as in B”, rather than “as in C”.

February 9, 2025

GENETICS-2024-307756

Genotype-by-environment interactions shape ubiquitin-proteasome system activity

Dear Dr. Avery:

Two experts in the field have reviewed your manuscript, and I have read it as well. I am pleased to inform you that, with minor revisions, it is potentially suitable for publication in GENETICS. The reviewers have comments and concerns that need to be addressed in a revised manuscript. You can read their reviews at the end of this email.

It is most important that you address the following in your resubmission: reorganize parts of the manuscript to increase clarity, readability, and transparency about some experimental details, as described by the reviewers. Please also be sure to correct the error with the link to the code in the Data Accessibility statement.

We look forward to receiving your revised manuscript. Please let the editorial office know approximately how long you expect to need for revisions.

Upon resubmission, please include:

1. A clean version of your manuscript;
2. A marked version of your manuscript in which you highlight significant revisions carried out in response to the major points raised by the editor/reviewers (track changes is acceptable if preferred);
3. A detailed response to the editor's/reviewers' comments and to the concerns listed above. Please reference line numbers in this response to aid the editors.

Additionally, please ensure that your resubmission is formatted for GENETICS.

<https://academic.oup.com/genetics/pages/general-instructions>

Follow this link to submit the revised manuscript: Link Not Available

Sincerely,

Patricia Wittkopp
Associate Editor
GENETICS

Approved by:
Anthony Long
Senior Editor
GENETICS

Reviewer #1 :

Gene-environment interactions (GxE) have widespread effects on molecular traits across species. In the present study, Avery, et al. have dissected the genetic architecture of the ubiquitin-proteasome system using an extensive design across two isolates of yeast and six UPS substrates/reporters in eight different relevant environments. They identified >400 QTLs across ~80 genomic regions associated with differences in UPS activity, many of which display substrate- and/or environment-specific effects. Using prior evidence, including some of their own studies, the authors nominate causal genes at several loci.

Overall, I am impressed with the solid study design and believe this will be of interest to multiple research communities. In general, I think additional interpretation of the results would improve the utility of the data set as a resource to motivate future studies. I have only minor comments:

I appreciate that the authors are able to replicate strong concordance between their previous QTL studies and the results reported here. It would further enhance the present work if the full data summarized in Supplementary Figure 2E and F were included as Supplementary Tables or as an additional column in Supplementary Table 4 denoting whether the QTL replicates or is novel, where applicable.

It seems like you see more cases of presence/absence GxE where the QTL is absent in SC and appears in a new environment

vs. where the QTL is present in SC and disappears in a new environment. Do you have any interpretation of what's biologically happening at loci where the QTL disappears in a new environment?

The authors report several candidate causal genes underlying a number of QTLs, either with previously published studies or by identifying genes with reasonable/relevant functions. While noting that I'm aware of the extensive amount of effort put into this study as it stands, my enthusiasm would be greatly increased by a bit more mechanistic work. In a previous article (M.A. Collins, et al. 2022), you report using CRISPR allele-swapping to investigate RM vs. BY alleles at the UBR1 locus. Are these strains still available for you to replicate the GxE results here and/or further investigate how these environmental conditions are modulating genetic effects at this locus?

Reviewer #2 :

This paper investigates the effect of different environments on genetic variation controlling ubiquitin-proteasome system (UPS) activity in *Saccharomyces* yeast. To quantify environmental and genotype-environment interactions, the authors use a previously developed high-throughput UPS-dual reporter system engineered into the genomes of a panel of recombinant genotypes made from crossing BY and RM strains. The UPS-activity reporter constructs consist of a fused UPS-target motif, a slow-maturing fluorophore (RFP), and a fast-maturing fluorophore (GFP), and this different rate of maturation allows inference of protein age from relative GFP/RFP signal and estimation of UPS activity. Using this system with bulk-segregant mapping strategies, the authors identify QTLs that cause variation in UPS activity against different substrates and in different environments. The results both recapitulate past findings about the remarkable amount of segregating variation in UPS activity and provide a significant advance by showing that the effects of those genetic variants are frequently environment-dependent and mediated through both direct components and indirect regulators of UPS.

The question investigated is central to better understanding gene regulation and the causes of its natural variation, the approach is elegant and powerful, and the claims are generally well-supported by the results. There are some ambiguous areas where more details/clarification are would improve the manuscript, highlighted below:

1. Sufficient detail to understand the system.

This paper naturally builds off of the foundation laid in the groups' past work on the UPS, but the essential details of the system should still be provided in this manuscript. Providing this background will make certain parts of the paper less confusing. Some specific areas:

1.1 The authors state the core logic of the system (Lines 169-171) but one more sentence unpacking it and/or giving an example of how the data are to be interpreted would make the downstream results easier to understand. Further, the authors use mCherry and mRuby for different targets (~40 min vs ~168 min maturation time). This should not affect any of the key claims that follow, but the authors should briefly articulate why this is the case.

1.2 The paragraph starting at 198 can be confusing if one lacks knowledge of the ubiquitin system and how that's different from just processes involving ubiquitin. For example, the authors state "the second ubiquitin system-independent reporter contains a linear fusion of four ubiquitin molecules..." This is technically correct, as the fused protein does not require the E1/E2/E3 upstream pathway but the authors should briefly explain why a protein with 4 ubiquitins fused on the N-term is a qualitatively different target from the ones described in the paragraph above.

1.3 Figures 2A and 2B show the difference of UPS activity from baseline but not the actual values. Those values, which I found to be very helpful in evaluating the claims, are shown in Supplementary Figure 1A. I would recommend the authors include this plot as a panel in the main text figure. Further, the authors need to include some measure of variation (e.g, sd, 95% CI) and information about the replicates in the figure legends, here and elsewhere.

2. Recommendation about organizing the results.

The authors present numerous results that are noteworthy and speak to the power of this system. However, the results section beginning on line 280 is currently 10 paragraphs and could be further shortened, divided, and/or reorganized to improve clarity and better highlight the findings. Doing so would make it easier to appreciate the salient points. For example, the authors state 1) "Across pathways, most QTLs were unique for a given substrate" (line 340), "Many QTLs mapped to the same genomic locations" (Line 357), and 3) "QTLs were not specific to a single environment" (Line 376). All of these are important, well-supported results, and structuring the manuscript to emphasize how they can all be true and how each result speaks to different questions about environment and trait architecture will benefit the paper.

Other comments:

Line 333: Clarify what is meant by "physiologically relevant categories."

Lines 350-353: The authors state that as wild yeasts often have to endure nutrient deprived conditions, RM may harbor more

genetic variation affecting UPS activity in starvation conditions than BY. It is not entirely clear whether they are referring to genetic variation within RM (which can't affect their observations) or variation between BY and RM. An alternative phrasing that would help clarify is that it is possible that relaxed selection in BY's lab environment led to a loss of BY alleles associated with higher UPS activity in starvation conditions.

There are several instances in which the authors suggest possible causal genes to explain their QTL peaks. In some places this might be misleading if the peaks contain many genes - the approximate number of variants in these peaks should be reported in some form.

Line 413: The authors require sign-change GxE to only be seen in 1 of 2 replicates, even though their presence/absence GxE needed to be seen in both replicates. Why is there a different detection criteria?

Recommendation: The authors conclude by highlighting the complexity of UPS activity (lines 552-561) and how this "complexity underscores the challenge of predicting molecular and organismal phenotypes from genotype." This is undoubtedly true, but I believe that the authors use this very powerful system to show that some of this complexity in gene regulation may be more predictable, especially since the hot-spots for transcriptional regulation found in previous studies and for post-translational regulation (such as UPS activity) overlap (lines 527-542). I believe emphasizing this point will further highlight a key contribution of this paper individually and of the BY/RM mapping system in general.

Reviewer #3 :

I was unable to access the code through the link included in the Data Availability statement.

Associate Editor Comments:

Response to editor/reviewers by Randi R. Avery, Mahlon A. Collins, and Frank W. Albert for “Genotype-by-environment interactions shape ubiquitin-proteasome system activity.” Resubmitted June 2025

Note: Line numbers refer to the clean, revised version of the manuscript provided in the resubmission.

Editor’s Comment from Decision email

It is most important that you address the following in your resubmission: reorganize parts of the manuscript to increase clarity, readability, and transparency about some experimental details, as described by the reviewers. Please also be sure to correct the error with the link to the code in the Data Accessibility statement.

We thank the editor and the three reviewers for their positive and constructive comments. We have addressed all of the reviewer comments below. The revisions include reorganization of several Results sections, in part by moving text and in part by adding a series of subheadings to break up longer sections. We have also fixed the error linking to our analysis code, and apologize for missing this during our initial submission. Our detailed responses are below.

Reviewer #1 :

Gene-environment interactions (GxE) have widespread effects on molecular traits across species. In the present study, Avery, et al. have dissected the genetic architecture of the ubiquitin-proteasome system using an extensive design across two isolates of yeast and six UPS substrates/reporters in eight different relevant environments. They identified >400 QTLs across ~80 genomic regions associated with differences in UPS activity, many of which display substrate- and/or environment-specific effects. Using prior evidence, including some of their own studies, the authors nominate causal genes at several loci.

Overall, I am impressed with the solid study design and believe this will be of interest to multiple research communities. In general, I think additional interpretation of the results would improve the utility of the data set as a resource to motivate future studies. I have only minor comments:

We thank the reviewer for these positive comments. Please note that we have numbered the reviewer’s comments below.

1. I appreciate that the authors are able to replicate strong concordance between their previous QTL studies and the results reported here. It would further enhance the present work if the full data summarized in Supplementary Figure 2E and F were included as Supplementary Tables or as an additional column in Supplementary Table 4 denoting whether the QTL replicates or is novel, where applicable.

This information has been added to the updated Supplementary Table 4

2. It seems like you see more cases of presence/absence GxE where the QTL is absent in SC and appears in a new environment vs. where the QTL is present in SC and disappears in a new environment. Do you have any interpretation of what's biologically happening at loci where the QTL disappears in a new environment?

To address this excellent comment, we first counted how many QTLs appear and disappear in a non-SC condition, breaking the results down by environment and reporter. As the reviewer surmised, the majority (60%, 152 / 254) of the presence / absence QTLs “appear” in a new environment. This leaves 40% of presence / absence QTLs that disappear in the non-SC condition. These cases are found in all environments and all for all reporters, without obvious, biologically interpretable patterns (e.g., there are no consistent differences between ubiquitin system-dependent vs. -independent reporters).

Environment	Absent in Condition; Present in SC ("disappear")	Present in Condition; Absent in SC ("appear")
4NQO	11	16
AZC	21	21
BTZ	13	18
LiAc	12	10
Low G	23	51
Low N	17	25
YNB	5	11
Total	102	152

Reporter	Absent in Condition; Present in SC ("disappear")	Present in Condition; Absent in SC ("appear")
4xUb	5	16
Asn	6	41
Phe	40	17
Rpn4	14	30
Thr	26	30
UFD	11	18
Total	102	152

QTLs that “disappear” in specific non-SC conditions are indeed interesting, and could be caused by at least two phenomena. First, in the starvation conditions (low G, low N, and YNB), QTLs that affect processes that are only active in the richer SC medium may disappear. QTLs disappearing in conditions in which a compound was added to SC require other explanations. One possibility is that QTLs detected in SC primarily affect UPS activity towards regulating physiological protein abundance. In environments where protein misfolding and aggregation are prevalent, these QTLs may disappear when UPS activity is primarily directed towards quality control, which often involves distinct pathways of substrate targeting and degradation. There may also be technical reasons, such that the presence of other QTLs that appear in the new condition reduces the relative contribution of the SC-only QTLs, reducing the heritability contributed from those QTLs such that they are now longer impactful. Due to the speculative nature of these interpretations, we have chosen not to include them in the paper.

3. The authors report several candidate causal genes underlying a number of QTLs, either with previously published studies or by identifying genes with reasonable/relevant functions. While noting that I'm aware of the extensive amount of effort put into this study as it stands, my enthusiasm would be greatly increased by a bit more mechanistic work. In a previous article (M.A. Collins, et al. 2022), you report using CRISPR allele-swapping to investigate RM vs. BY alleles at the UBR1 locus. Are these strains still available for you to replicate the GxE results here and/or further investigate how these environmental conditions are modulating genetic effects at this locus?

We appreciate the reviewer’s point that the paper represents an “extensive amount of effort.” We certainly agree that exploration of the mechanisms that underlie GxE arising from specific DNA variants is an interesting question. However, due to the significant effort required to do this question justice, we prefer to leave it to future work.

Reviewer #2 :

This paper investigates the effect of different environments on genetic variation controlling ubiquitin-proteasome system (UPS) activity in *Saccharomyces* yeast. To quantify environmental and genotype-environment interactions, the authors use a previously developed high-throughput UPS-dual reporter system engineered into the genomes of a panel of recombinant genotypes made from crossing BY and RM strains. The UPS-activity reporter constructs consist of a fused UPS-target motif, a slow-maturing fluorophore (RFP), and a fast-maturing fluorophore (GFP), and this different rate of maturation allows inference of protein age from relative GFP/RFP signal and estimation of UPS activity. Using this system with bulk-segregant mapping strategies, the authors identify QTLs that cause variation in UPS activity against different substrates and in different environments. The results both recapitulate past findings about the remarkable amount of segregating variation in UPS activity and provide a significant advance by showing that the effects of those genetic variants are frequently environment-dependent and mediated through both direct components and indirect regulators of UPS.

The question investigated is central to better understanding gene regulation and the causes of its natural variation, the approach is elegant and powerful, and the claims are generally well-supported by the results. There are some ambiguous areas where more details/clarification are would improve the manuscript, highlighted below:

We thank the reviewer for this positive assessment, and have addressed their comments as described below.

1. Sufficient detail to understand the system.

This paper naturally builds off of the foundation laid in the groups' past work on the UPS, but the essential details of the system should still be provided in this manuscript. Providing this background will make certain parts of the paper less confusing. Some specific areas:

1.1 The authors state the core logic of the system (Lines 169-171) but one more sentence unpacking it and/or giving an example of how the data are to be interpreted would make the downstream results easier to understand.

We agree, and have added more explanation of the logic underlying TFTs to the main text (Lines 165-167).

To make the logic of the TFT system clearer, the revised manuscript includes an expanded schematic that illustrates how the TFT functions (Figure 1A). We have also provided an expanded explanation of the logic of the TFT system and a specific example that illustrates how the data are interpreted (Lines 174-180). With these additions, we believe that the results of the manuscript will be easier to understand.

Further, the authors use mCherry and mRuby for different targets (~40 min vs ~168 min maturation time). This should not affect any of the key claims that follow, but the authors should briefly articulate why this is the case.

In designing TFTs to measure UPS activity, a key consideration is the difference between the construct's degradation rate and the maturation rate of the RFP. A TFT will have good dynamic range for measuring differences in UPS activity when the maturation rate of its RFP is at least half that of the construct's degradation rate. Because the Thr Ac/N-degron's degradation rate is similar to mCherry's maturation rate (~40 minutes), a Thr Ac/N-degron mCherry / sfGFP TFT has limited ability to separate populations that differ in UPS activity towards this degron (Reviewer Response Fig. 1A). In contrast, the high and low UPS activity populations are well-separated by the Thr Ac/N-degron mRuby / sfGFP TFT, owing to the longer maturation rate of mRuby (~168 minutes; Reviewer Response Figure 1B). The other degrons we tested have substantially shorter half-lives, and UPS activity differences towards these substrates can be sensitively detected with the mCherry / sfGFP TFT. Thus, to ensure that we could obtain sensitive, precise measurements of UPS activity towards the Thr Ac/N-end degron, we used the mRuby / sfGFP TFT. We have adjusted the description in the Methods section (Lines 619-620) to make this clearer.

The distinct RFPs used for the Thr Ac/N-degron and other substrates could create differences in results, however, we think this is unlikely. We note that we and others have obtained similar results with TFTs that contain distinct combinations of fluorophores. For example, here and in our previous works, we detect the QTL on chromosome VII containing *RPT6*, which is expected to influence UPS substrates broadly, with the same effect direction using mCherry / sfGFP and mRuby / sfGFP TFTs across multiple substrates. Therefore, as the reviewer suggests, using mCherry and mRuby in separate TFTs is unlikely to affect the manuscript's key claims.

Reviewer Response Fig. 1: TFT dynamic range. Segregant populations containing either the Thr Ac/N-degron TFTs with either mCherry or mRuby were used to isolate cells with extremely high or low UPS activity. **A.** Thr Ac/N-degron mCherry / sfGFP TFT. **B.** Thr Ac/N-degron mRuby / sfGFP TFT.

1.2 The paragraph starting at 198 can be confusing if one lacks knowledge of the ubiquitin system and how that's different from just processes involving ubiquitin. For example, the authors state "the second ubiquitin system-independent reporter contains a linear fusion of four ubiquitin molecules..." This is technically correct, as the fused protein does not require the E1/E2/E3 upstream pathway but the authors should briefly explain why a protein with 4 ubiquitins fused on the N-term is a qualitatively different target from the ones described in the paragraph above.

We have substantially revised our description of the ubiquitin system-dependent and -independent reporters used in our study to improve clarity. Specifically, the revised text emphasizes that the 4xUb reporter provides a proteasome recognition element that is similar to that found on most endogenous UPS substrates, but that is produced without the involvement of ubiquitin system enzymes (Lines 207-208).

1.3 Figures 2A and 2B show the difference of UPS activity from baseline but not the actual values. Those values, which I found to be very helpful in evaluating the claims, are shown in Supplementary Figure 1A. I would recommend the authors include this plot as a panel in the main text figure.

We have added Supplementary Figure 1A as a new panel A in Figure 2, in addition to a new panel 2B that shows the same information but arranged by environment.

1.4 Further, the authors need to include some measure of variation (e.g, sd, 95% CI) and information about the replicates in the figure legends, here and elsewhere.

We appreciate the reviewer's request for transparent reporting of these results and would like to stress that the (previous) panels 2A & B are meant to be summaries of the data rather than full representations of all raw data. Figure panels 2D-F (now panels F – H) show the full raw data for three examples, in addition to boxplots. Plots in the same format showing the raw data for **all** the data underlying Figure 2 were (and still are) provided as Supplementary File 2. Further, the former panels 2A & B (now C & D) have a visual illustration of significance via the transparency of the points and lines.

To further increase the transparency of the reported results, we have made the following changes:

- Standard deviations are now shown in the new panels A and B in Figure 2.
- The number of biological replicates (eight) was added to the legend of Figure 2A & B, in addition to their previous inclusion in the Results section, the Methods, and the figure legend to the (previous) panels 2 D-F (now F-H).
- Standard deviations have been added to Supplementary Table 3.
- The number of replicates was added to the legend of Figure 1C.

- The number of analyzed QTLs is now given in the legend for panel F of the former Supplementary Fig. 2 (now S1), in addition to the existing breakdown of QTL numbers in the table in panel E.

2. Recommendation about organizing the results.

The authors present numerous results that are noteworthy and speak to the power of this system. However, the results section beginning on line 280 is currently 10 paragraphs and could be further shortened, divided, and/or reorganized to improve clarity and better highlight the findings. Doing so would make it easier to appreciate the salient points. For example, the authors state 1) "Across pathways, most QTLs were unique for a given substrate" (line 340), "Many QTLs mapped to the same genomic locations" (Line 357), and 3) "QTLs were not specific to a single environment" (Line 376). All of these are important, well-supported results, and structuring the manuscript to emphasize how they can all be true and how each result speaks to different questions about environment and trait architecture will benefit the paper.

We agree that this and other Results sections were too long and combined too many different kinds of results. We have made the following changes:

- We have reorganized and split the section pointed out by the reviewer into four sections. The first of these sections briefly introduces the QTL mapping strategy and then describes only results in SC medium, specifically those on reproducibility and QTLs found for the two new reporters (UFD and 4xUb). We then report QTLs in different environments, including the comparison of ubiquitin system-dependent and -independent UPS pathways, and between starvation and non-starvation conditions. Next, we describe our results showing that QTLs tend to occur at certain positions. Finally, we describe the loci whose effects are the most specific to a single environment.

- We have split the final Results section in two, by first presenting the result that half of all UPS activity QTLs show some form of GxE, followed by the section on direct vs. indirect effects.

- We have also restructured the first of these (now) two sections, so that the result for each GxE type now appears right after its definition. This change avoids the previous arrangement in which a long paragraph provided formal definitions of all the GxE types before the results were given in the next paragraph.

Other comments:

Line 333: Clarify what is meant by "physiologically relevant categories."

We have reworded this sentence as follows: “To examine these architectures in more detail, we collapsed combinations of reporters and environments into major UPS pathways and starvation vs. non-starvation environments.” (Lines 338-339)

Lines 350-353: The authors state that as wild yeasts often have to endure nutrient deprived conditions, RM may harbor more genetic variation affecting UPS activity in starvation conditions than BY. It is not entirely clear whether they are referring to genetic variation within RM (which can't affect their observations) or variation between BY and RM. An alternative phrasing that would help clarify is that it is possible that relaxed selection in BY's lab environment led to a loss of BY alleles associated with higher UPS activity in starvation conditions.

As RM is a haploid strain, it cannot contain variation within itself – it can only differ from other strains. We see how our wording in this section, which grouped RM with other wine strains, could have been interpreted otherwise, and have rephrased it. We have also added the reviewer's point about BY having potentially experienced relaxed constraints to the same section.

There are several instances in which the authors suggest possible causal genes to explain their QTL peaks. In some places this might be misleading if the peaks contain many genes - the approximate number of variants in these peaks should be reported in some form.

We have added the following sentence to the end of our brief description of the QTL mapping strategy acknowledging that all QTLs identified several genes. “All QTLs included multiple genes (Supplementary Table 4); likely candidate genes are presented throughout the text.” (Lines 291-293)

In addition, we have updated Supplementary Table 4 with the number of BY / RM variants and a list of the ORFs within each QTL.

Line 413: The authors require sign-change GxE to only be seen in 1 of 2 replicates, even though their presence/absence GxE needed to be seen in both replicates. Why is there a different detection criteria?

Our reasoning behind this choice of criteria is as follows:

The detection of loci with presence / absence GxE is complicated by false negatives, which occur when a QTL is missed due to incomplete statistical power. If a QTL is seen in only one replicate when testing for presence / absence, a parsimonious explanation is that it was missed in the second replicate due to incomplete power. Such a pattern is therefore not a strong

indication of GxE. To counter this, we used conservative criteria for calling presence / absence by requiring that 1) the QTL be present in both replicates of one condition and 2) there be no QTL in both replicates of the other condition.

False negative QTLs are less of an issue for sign changes because this form of GxE can only be called when QTLs have been detected in both conditions. We still require both replicates of one of the two environments to show QTLs with the same sign to be considered in these analyses. We then call a sign change if a QTL is detected in at least one of the replicates in the other condition. We acknowledge that this criterion is perhaps more lenient than that required for presence / absence, but given that sign changes were relatively rare in our data, we considered this an acceptable trade-off between rigor and discovery.

We stress that of the 17 sign changes we detected, only four are based on one (rather than two) replicates in the “other” condition. This information is included in Supplementary Table 6 under columns “QTL_1_num_of_replicates” and “QTL_2_num_of_replicates.” Further, at each of these four sign change loci, the “other” condition had a QTL with opposite sign in one replicate and no QTL in the other replicate. There were no cases in which the two replicates in the “other” condition both had a QTL but with opposite sign *within* this condition – an arrangement that is theoretically possible given our search strategy but one we did not encounter. We have added this breakdown of the 17 sign change loci to the Results section (Lines 437-439). We have also expanded our explanation of the choice to allow sign change detection based on one replicate in the second condition in the Methods (Lines 915-921).

Recommendation: The authors conclude by highlighting the complexity of UPS activity (lines 552-561) and how this "complexity underscores the challenge of predicting molecular and organismal phenotypes from genotype." This is undoubtedly true, but I believe that the authors use this very powerful system to show that some of this complexity in gene regulation may be more predictable, especially since the hot-spots for transcriptional regulation found in previous studies and for post-translational regulation (such as UPS activity) overlap (lines 527-542). I believe emphasizing this point will further highlight a key contribution of this paper individually and of the BY/RM mapping system in general.

We appreciate and agree with this recommendation, and have added the following sentence to the final conclusion paragraph: “At the same time, our finding that hotspots of transcriptional variation found in previous studies overlap with those that shape post-translational processes represented by UPS activity suggests that there is a shared, tractable set of causal genes and variants whose identification may enable prediction of the effects of genetic variation across the gene regulatory cascade.”

Reviewer #3 :

Comment enclosed in decision email:

I was unable to access the code through the link included in the Data Availability statement.

We apologize for this oversight, which has now been fixed. All code is available at https://github.com/randiraphd/gxe_ups_activity

Comments that were enclosed as pdf (note that we have numbered the reviewer comments):

Summary:

The authors measure the activity of the yeast ubiquitin-proteasome system (UPS) in seven environments, using six fluorescent reporters – two of which have been newly designed for this study. Most of these environments are stresses which are known to affect UPS activity in defined ways, yet many discordant effects are observed between strains BY and RM. This thus reveals genotype-by-environment (GxE) interactions for UPS activity. These interactions are next investigated through QTL mapping, which shows that variation is mostly found at a limited number of loci. Some contain genes which are directly involved in the UPS, while others interestingly contain genes known as trans regulatory hotspots, and even genes involved in response to some of the stressors. More importantly, the authors investigate how GxE arise from these QTLs, which further emphasizes that variation among strains and environments is caused by a small number of loci, and that not all are directly related to the UPS. This work reveals an underappreciated potential for GxE at the level of protein degradation, while also highlighting how these interactions may vary across UPS pathways. This has many interesting implications, for instance regarding the evolution of protein abundance.

Strengths :

The experimental approach developed by this group and used in the current study is innovative and particularly interesting. Applying it to GxE seems like a logical and important next step after their previous papers. The tandem fluorescent protein timers used should especially be more widely known. The work is very rigorous, with biological replicates throughout and the use of a common starter culture to inoculate the different environments. The dissection of GxE into three types at the level of individual loci provides valuable insights into the different mechanisms by which variation in protein degradation can arise. Some additional analyses could however potentially help in making them clearer for readers (see below).

We thank the reviewer for these positive comments!

Major comments :

1. I have not been able to access the code related to this work. The Github link in the Data availability section leads to an empty repository. This is certainly not intentional from the authors, but it has nonetheless limited my understanding of the analyses which have been performed.

We apologize for this oversight, which has now been fixed. All code is available at https://github.com/randiraphd/gxe_ups_activity

2. My main concern about the paper is that it may be slightly too descriptive. I found it difficult to summarize the findings into a few “take-home” messages, which also made it harder to follow the reasoning from one section to the next. It is however entirely clear to me that the authors did very interesting and relevant work. This data seems to provide lots of insights into how variation in protein degradation can emerge, but they are not always made clear to the readers. I would especially be interested in more comparative analyses of the properties of loci which give rise to QTLs with and without GxE. I’m under the impression that adding some more summary figures and performing additional such analyses would make the paper more approachable for readers. I hope that my suggestions will prove helpful, but I am aware that the authors are in a much better position than me to assess their relevance.

- In general, it seems to me that the fact that there are both ubiquitin-dependent and ubiquitin-independent reporters, which give complementary information on the UPS, was not emphasized enough throughout the results. Since so much data is shown, it’s easy for the reader to forget this information (and its significance), especially as not all figure panels make it clear.

To make this distinction (as well as those between starvation and non-starvation conditions) more visible for the reader, we have updated the Figures as follows:

– Figure 2 now shows the pathway and environment types on top of the (new) panels A and B, as well as in the legends to panels A – D.

– In Figure 3B (previously 3D), we added annotations to the environments and pathways.

With these additions, the distinction between pathway types and environments is now clearly visible in each major display item.

- Figure 2 could probably benefit from one (or two) figure(s) summarizing the frequency and direction of effects, across reporters and environments. I found panels A and B slightly difficult to interpret, and replacing one of them with a figure showing mean effects and occurrences of positive and negative changes might make it easier for readers to notice the trends in the data.

Similarly, Supp Fig 1B-C show how GxE frequency varies between reporters and environments, which seems important enough to be included in a main figure.

We point out that the heatmap shown in panel 2C (now 2E) does provide a summary of all the results very similar to what the reviewer suggests, including directions and magnitude of GxE effects as well as whether each comparison showed significant GxE. In light of this, we have chosen to remove panels S1 B&C, since the heatmap in panel 2E already shows all this information in a compact manner.

To further increase the clarity of our data display, we have added new panels 2A & B to make it easier to understand what is shown in panels A & B of the initial submission (now panels C & D).

- Figure 3, and especially panel A, is striking, but I wonder if it could benefit from a panel showing how the number of QTLs detected is related to the magnitude of environmental effects seen in Fig 2. I'm under the impression that this would help connect these two sections of the manuscript. Similarly, are there more or less QTLs when a GxE is observed between BY and RM on Fig 2?

We thank the reviewer for these suggestions. To explore the relationship between the parental results and the QTL maps, we conducted two analyses.

First, we compared the magnitude of the difference in UPS activity between the BY and RM strains for a given reporter in a given environment to the number of QTLs found in the same reporter / environment combination (Reviewer Response Fig. 2). While the resulting correlation coefficient was indeed positive ($r = 0.23$), the correlation was not statistically significant ($p = 0.116$):

Reviewer Response Fig. 2: Correlation of difference in UPS activity between BY and RM and total QTLs per reporter / environment combination.

Dividing the results by reporter and environment did not reveal clear patterns (Reviewer Response Fig. 3):

Reviewer Response Fig. 3: Correlation of difference in UPS activity between BY and RM and total QTLs per reporter / environment combination. Data as in Reviewer Response Fig. 2, but with reporter and environment specified.

Regarding the reviewer's second suggestion, we divided the experiments (i.e., combinations of reporter and environmental contrasts to SC) according to whether they did or did not have significant GxE in the parents. We then compared the number of QTLs with and without GxE (Reviewer Response Fig. 4). There was no significant difference for either group:

Reviewer Response Fig. 4: Comparison of reporter / environment combination that had GxE between BY and RM and the total number QTLs with and without GxE. **A.** Comparison of QTLs with GxE. **B.** Comparison of QTLs that did not show GxE.

We then used the counts above to compute the fraction of QTLs with GxE for comparisons with and without GxE (Reviewer Response Fig. 5). We again found no difference:

Reviewer Response Fig. 5: The fraction of QTLs with GxE per reporter / environment combination and whether or not the same reporter / environment combination had GxE between BY and RM.

Collectively, these analyses show that neither the strength of the parental difference in UPS activity nor the presence of GxE in the parental comparisons is predictive of the underlying QTL patterns. We caution that these analyses come with the considerable caveat that we are unable to meaningfully consider the strength of the QTLs. A single QTL of large effect could underlie most of a given parental difference or GxE pattern, with much weaker contributions from small-effect QTLs. Such a pattern would not be detectable with the analyses above. Unfortunately, the bulk segregant approach used here, although statistically powerful, does not produce easily-interpretable effect sizes that could be summed across QTLs into a joint, additive effect and then compared to a parental difference as in association-based mapping approaches. While a QTL producing a large effect is expected to produce a larger allele frequency difference in bulk segregant QTL mapping, the relationship between QTL effect and allele frequency difference is not linear and can only be approximated using highly simplified assumptions. Further, given the data is produced via competitive cell sorting in mixed cell populations, the presence of strong QTLs could downwardly bias the observed effect size of other QTLs (intuitively, this can happen if a strong QTL dominates the among-cell distribution such that weaker QTLs do not influence whether a cell is sorted into the high or low pool in a way that linearly scales with the effects of the weaker QTLs). Any epistasis between QTLs is also undetectable in these data and could skew the observed allele frequency differences. Together, these reasons render the analyses above inconclusive, and we therefore prefer not to include them in the paper.

- Fig 5A is especially interesting, but I am unsure about the relevance of the corresponding panel B. Since the pairs showing GxE are a subset of the total pairs, and I don't understand how this correlation is informative. Perhaps, the saturation effect which is visible on the scatterplot is what should be emphasized? Otherwise, it might be worthwhile to replace this panel with another one showing the percentage of QTL bins showing GxE which also contain UPS-associated genes. This analysis has been performed for sign change GxE (lines 477-480), but I wonder why it is not also discussed for "presence/absence" and "no GxE" cases.

We thank the reviewer for their positive comment on Figure 5A and note that during this revision, we have updated the text describing this figure panel. Specifically, we now describe the top six (rather than top five) bins with the most cases of GxE, as we noticed that the bin containing *RPT6* is tied with another bin without an obvious candidate gene. We have also emphasized our observation that the *RPT6* locus is unusual among the top bins with likely candidate genes in that it has only presence / absence GxE but no sign changes.

We have chosen to keep panel 5B, as we believe that it makes an important point, perhaps mostly in what it does not show: GxE does not occur at a limited, privileged set of QTLs that are environmentally responsive but instead is a common feature of loci that have broad effects. We have attempted to clarify this point in the Results describing panel 5B.

Regarding the reviewer's suggestion for a more systematic analysis of potential causal genes in regions with sign changes compared to regions with presence / absence GxE, we want to point out that the current final Results section on sign changes was done "by hand." This was

possible given there were only 29 QTLs involved in the sign change pairs, such that their confidence intervals could be plotted and examined manually in the UCSC genome browser. A similar analysis of the much larger number of cases with presence / absence GxE would not be feasible. Further, our goal in this section was not to draw a contrast between sign changes and presence / absence but a focused exploration of the sign change regions. Of course, all QTLs identified in this study are available with this paper as Supplementary Table 4, such that interested readers can perform additional analyses.

I may be overstepping my role as a reviewer with the suggestions above, and I apologize if that is the case. My general impression when reading the manuscript was that the figures may be too focused on showing the data itself instead of what should be concluded from it. This made it harder for me to understand and follow the paper, but I might be part of a very small minority of readers in that regard.

We thank the reviewer for their close engagement with our paper, and greatly appreciate their thoughtful suggestions!

3. An additional concern I have is about the replicability of previously detected QTLs (lines 292-301). While I trust the authors, I find that this part of the paper could be more convincing. My understanding is that only the QTLs which have been identified in the present study but not reported in the two previous studies are discussed. I suppose that any QTLs which were detected in the previous studies but not replicated in this one also have lower LOD scores and effect sizes, but I think it would be important to mention it as well.

To address the reviewer's concern, we examined reproducibility of the previously published QTLs in the QTLs identified here. Briefly, replication rates were very similar to those we had obtained testing the QTLs identified here in earlier data.

In more detail, there were a total of 31 QTLs for the Asn, Phe, Rpn4, and Thr reporters in SC medium in Collins et al. 2022 and 2023 (see the Reviewer Table below). Of these, 24 (77%) replicated in the data reported here – the same fraction we observed when assessing replication of the QTLs mapped here in earlier data.

Reporter	QTLs present in both studies	QTLs in Collins et al., 2022, 2023	Percent Replicated
Asn	4	7	57
Phe	9	10	90
Rpn4	2	5	40
Thr	9	9	100
Total	24	31	77

As observed in the initial manuscript, published QTLs that did not replicate in our new data had smaller LOD scores (although this difference was not statistically different in this case: $p = 0.072$) and smaller allele frequency differences ($p = 0.033$) than QTLs that did replicate in our new data (Reviewer Response Fig. 6). These results are consistent with our earlier results in the manuscript as well as our general experience with this bulk segregant mapping technique: reproducibility is excellent for stronger loci, while weaker loci can sometimes be missed due to incomplete power. We have added a statement about these results to the manuscript (Lines 302-304).

Reviewer Response Fig. 6: LOD (left) and effect sizes (right) of QTLs mapped in Collins et al., 2022 and 2023 that did not (left boxplots) and did (right boxplots) replicate in the current QTL data. Mean values were: LOD: Did not replicate = 15.9; Replicated = 38.5; ΔAF : Did not replicate = 0.152; Replicated = 0.233.

Minor comments :

4. The introduction nicely exposes both the UPS system and the variability of protein degradation. I however wonder whether the paragraphs discussing GxE (lines 31-69) could benefit from being shortened a little. It seems to me that some of these examples distract a bit from the main point of the paper. This is however highly subjective and I may be completely wrong in this regard.

We thank the reviewer for this suggestion. Our goal in these two paragraphs was to provide a view of the GxE literature across species that was purposefully wider than what may be strictly necessary, to both orient the reader and credit a wider range of prior work. As such, we have decided to keep this Introduction section as is.

5. In general, I think there may be a bit too much back-and-forth between figures and panels throughout the results section, which makes it harder to follow:

- Fig 1A is first referenced at line 139, but it is not fully explained until line 291. This confused me at first, although the detailed explanation of the experimental approach at lines 281-291 was crystal clear.

We see the reviewer's point that while most of Figure panel 1A is described in this first section of the Results, the QTL mapping cartoon at the bottom right of the panel was indeed not referred to until much later. To avoid this confusion, we have now added a brief mention (underlined here) of genetic mapping to the first Results sentence: "To study GxE in the UPS, we compared the UPS activity of two genetically divergent yeast strains towards six UPS substrates that engage multiple distinct UPS pathways in eight environments including multiple starvation and chemical stressors, followed by genetic mapping of these UPS substrates in all eight environments (Fig. 1A)." (Lines 136-137)

- In the next section (lines 222-275), Fig 2A-C is first referenced (lines 232, 238). Then, Fig 2D (line 240, 2E (line 244) and 2F (line 248) are discussed in order. After this however, the text goes back to Figs 2A, B & C, which are referenced multiple times (lines 252, 255, 258, 263 and 275). A way to limit this back-and-forth could be to first fully describe the environmental effects on UPS activity, which is currently split between lines 225-233 and lines 263-268, before describing the significant GxE detected.

To address this, we have streamlined the figure callouts in this section but have also retained some callouts to earlier-cited panels. Across its different panels, Figure 2 shows UPS activity (the new panels A & B), environment effects obtained by subtracting UPS activities in a given environment from those in SC (C & D), and a summary of all results in panel E. Panels A – E therefore all show the same data at different levels of abstraction. In principle, they could be

called out collectively throughout this section. However, specific statements in this Results section are more vividly illustrated by some of these panels than others. To help guide the reader, we feel it is appropriate to refer back to those panels even if they have been called out earlier.

We have chosen to keep the organization of this section unchanged. It appears to us to be most logical to first describe results on environment effects per strain, first *without* using the linear models to formally test for GxE. This is information presented in the first paragraph of this section. The fact that there are cases where one strain had a significant environment effect while the other did not then leads naturally to the question of GxE, which requires the linear model including an interaction term. The rest of the section then describes examples and patterns of significant cases of GxE as identified by significant interaction terms (those marked by opaque lines in panels 2C & D, and by points in the cells of the heatmap in 2E).

We acknowledge that it may not have been clear that the entire remainder of this section after the linear models refers only to cases with significant GxE. We have rewritten and simplified this section slightly to make this clearer. First, we have added the following sentence after the linear models have been introduced: "Specific examples and patterns among cases with significant GxE are presented below." Second, we have reworded the final paragraph of this section and removed most p-values that were given. These p-values were for per-parent T-tests comparing SC to the given condition, but may have created the impression that they were for the interaction (i.e., GxE) term in the linear model. All p-values for all parental tests do of course remain available in Supplementary Table 3, and significant tests remain visible as opaque points in Figure 2C & D.

- In addition, although this is not related to the organization of the results section, one thing I found slightly confusing about Fig 2A-B is that I was not totally sure whether the baseline was the same for BY and RM, or rather strain-specific.

These baselines were strain-specific. The legend to Figure 2 and the Methods were updated to clarify this.

- The same kind of back-and-forth between panels occurs for Fig 3. The first panel to be referenced individually is 3D (line 308), later followed by 3B (lines 309, 313, 335), 3C (line 344) and then 3A (line 357).

This has been streamlined, in part by moving the previous panel 3D into position 3B.

6. I may be mistaken, but it seems that the 4NQO abbreviation is introduced at line 154 without being first defined.

This oversight has been corrected in the revised manuscript.

7. As a final (very) minor comment, there seems to be a mistake in the legend of Supplementary Fig 1C. It should probably read “Data as in B”, rather than “as in C”.

Thank you for this correction. Based on other comments we removed Supplementary Figure 1.

July 21, 2025

RE: GENETICS-2025-308286

Dr. Randi Rae Avery
University of Minnesota Twin Cities
Genetics, Cell Biology, and Development
420 Washington Ave SE
Minneapolis, Minnesota

Dear Dr. Avery:

Congratulations, your manuscript titled "Genotype-by-environment interactions shape ubiquitin-proteasome system activity" is accepted for publication in GENETICS! Many thanks for submitting your research to the journal.

The reviewers had a few suggestions for improving the manuscript that you may want to consider. You can view their comments at the bottom of this email. Please note that the need to include flow cytometry data noted by reviewer 2 is required prior to publication.

To Proceed to Publication:

1. Format your article according to GENETICS style: <https://academic.oup.com/genetics/pages/general-instructions>

2. Ensure that you comply with data and community resource citation guidelines:
<https://academic.oup.com/genetics/pages/general-instructions#Data-Policy>

3. Upload your final files at <https://genetics.msubmit.net>

4. Add oupsupport@scipris.com and genetics.oup@novatechset.com (or the domains @scipris.com and @novatechset.com) to your email program's "safe senders" list. You will be contacted by both at various points during the production process.

Notes:

- Your currently-accepted manuscript (unedited, as submitted, reviewed, and accepted) will be published at GENETICS and deposited into PubMed as an Advance Access article. Notify sourcefiles@thegsajournals.org before signing your license if you do not wish to publish your article via Advance Access.

- We invite you to submit an original color figure related to your paper for consideration as cover art. Please email your submission to the editorial office or upload it with your final files. You can submit a small-sized image for evaluation, and if selected, the final image must be a TIFF file 2513px wide by 3263px high (8.375 by 10.875 inches; resolution of 600ppi). Please avoid graphs and small type.

- After files are sent to Oxford University Press we use SciPris to manage article licensing and payment. If you do not have a SciPris account, you will receive an email from no-reply@scipris.com to sign up to use Oxford University Press' author portal. After logging in, follow the online instructions to sign your license and arrange any payment due.

If you have any questions or encounter any problems while uploading your accepted manuscript files, please email the editorial office at sourcefiles@thegsajournals.org.

Sincerely,

Patricia Wittkopp
Associate Editor
GENETICS

Approved by:
Anthony Long
Senior Editor
GENETICS

Review comments (if applicable):

Reviewer #2 :

The authors have provided well-reasoned, thoughtful revisions and responses to all of the issues raised previously. I found the modifications to Figure 1 to be particularly effective in communicating key details about the system and the new sectioning of the Results section to be effective in highlighting the different discoveries. I have no further comments.

Reviewer #3 :

I have only few comments about this revised version of the manuscript. With the streamlining of the Results section which has been done, all concerns (sometimes misguided) I had about the paper's clarity and focus are now irrelevant. Similarly, I initially failed to understand the significance of some observations, which the authors now very clearly emphasize (especially about Fig 5B; lines 474 and 532-535). Overall, this is very interesting work which makes important observations about the genetic architecture and evolution of protein decay rates, while showing the power of the combination of TFTs with bulk segregant analysis.

I found the authors' additional analyses, about the number of QTLs and the magnitude of environmental effects as well as the presence or not of GxE between BY and RM, to be interesting. I understand their decision not to include them in the paper, since they are inconclusive. These analyses nonetheless emphasize that it is not as simple as higher numbers of QTLs necessarily resulting in greater between-strain differences and likelier GxE.

In addition to two very minor comments (see below), I have one (minor) concern, which I should probably have included in the first review. Unless I am mistaken, the raw data for the cytometry of the parental BY and RM strains is included neither as Supplementary nor as part of the Github repository. Especially since no supplementary figures display the underlying distributions from which mean UPS activity is obtained, I think that this data should be made available.

My two minor comments are the following:

- Line 32: I think a comma is missing between "phenylketonuria" and "individuals".
- Line 266: Similarly, a comma may be missing between "LiAc" and "GxE".

These seem to me like typos which should be corrected, but I may be wrong, as I am not a native English speaker.